# Boundary Dpp promotes growth of medial and lateral regions of the *Drosophila* wing

**Lara Barrio[1,3], Marco Milán[1,2,3]\***

[1]Institute for Research in Biomedicine (IRB Barcelona), Barcelona, Spain; [2]Institució Catalana de Recerca i Estudis Avançats (ICREA), Barcelona, Spain; [3]The Barcelona Institute of Science and Technology (BIST), Barcelona, Spain

**Abstract** The gradient of Decapentaplegic (Dpp) in the *Drosophila* wing has served as a paradigm to characterize the role of morphogens in regulating patterning. However, the role of this gradient in regulating tissue size is a topic of intense debate as proliferative growth is homogenous. Here, we combined the Gal4/UAS system and a temperature-sensitive Gal80 molecule to induce RNAi-mediated depletion of *dpp* and characterise the spatial and temporal requirement of Dpp in promoting growth. We show that Dpp emanating from the AP compartment boundary is required throughout development to promote growth by regulating cell proliferation and tissue size. Dpp regulates growth and proliferation rates equally in central and lateral regions of the developing wing appendage and reduced levels of Dpp affects similarly the width and length of the resulting wing. We also present evidence supporting the proposal that graded activity of Dpp is not an absolute requirement for wing growth.

**\*For correspondence:** marco. milan@irbbarcelona.org

**Competing interests:** The authors declare that no competing interests exist.

## Introduction

Decapentaplegic (Dpp), the *Drosophila* BMP homolog, has served as a paradigm to characterize the role of morphogens in regulating patterning of developing tissues (*Affolter and Basler, 2007*; *Restrepo et al., 2014*). In the developing *Drosophila* wing disc, Dpp is expressed in a central stripe that corresponds to the anterior-posterior (AP) compartment boundary, and its gradient provides a series of concentration thresholds throughout the tissue that set the transcriptional state of target genes in discrete domains of gene expression as a function of their distance from the source (*Lecuit et al., 1996*; *Nellen et al., 1996*). Graded activation of the Dpp transducer MAD and the inverse gradient of Brinker, a transcriptional repressor negatively regulated by Dpp, contribute to the transcriptional regulation of Dpp target genes in discrete domains (*Affolter and Basler, 2007*; *Restrepo et al., 2014*). These domains are ultimately used to locate the patterning elements of the adult wing [e.g. longitudinal veins, (*de Celis et al., 1996*; *Sturtevant et al., 1997*)]. Thus, the participation of the Dpp gradient in specifying cell identities in a concentration-dependent manner is well accepted.

By contrast, the role of the Dpp gradient in regulating tissue size has been a topic of intense debate as proliferative growth is homogenous in the developing wing (*Milán et al., 1996*). The classical 'steepness model' proposes that the juxtaposition of cells sensing disparate levels of the morphogen promotes proliferative growth (*Lawrence and Struhl, 1996*; *Rogulja and Irvine, 2005*). This model is questioned by the observation that the Dpp-gradient scales with the size of the wing primordium and that the slope of the gradient does not change (*Wartlick et al., 2011*). These same authors propose a new model - the 'temporal rule model' - that posits that cells divide when Dpp signaling levels have increased by 50%. However, cells lacking MAD and Brinker activities - in which

**eLife digest** From the wings of a butterfly to the fingers of a human hand, living tissues often have complex and intricate patterns. Developmental biologists have long been fascinated by the signals – called morphogens – that guide how these kinds of pattern develop. Morphogens are substances that are produced by groups of cells and spread to the rest of the tissue to form a gradient. Depending on where they sit along this gradient, cells in the tissue activate different sets of genes, and the resulting pattern of gene activity ultimately defines the position of the different parts of the tissue.

Decades worth of studies into how limbs develop in animals from mice to fruit flies have revealed common principles of morphogen gradients that regulate the development of tissue patterns. Morphogens have been shown to help regulate the growth of tissues in a number of different animals as well. However, how the morphogens regulate tissue size and what role their gradients play in this process remain topics of intense debate in the field of developmental biology.

In the developing wing of a fruit fly, a morphogen called Dpp is expressed in a thin stripe located in the center and spreads to the rest of the tissue to form a gradient. Barrio and Milán have now characterized where and when the Dpp morphogen must be produced to regulate both the final size of the fly's wing and the number of cells the wing eventually contains. The experiments involved preventing the production of Dpp in the developing wing in specific cells and at specific stages of development. This approach confirmed that Dpp must be produced in the central stripe for the wing to grow. Matsuda and Affolter and, independently, Bosch, Ziukaite, Alexandre et al. report the same findings in two related studies. Moreover, Barrio and Milán and Bosch et al. also conclude that the gradient of Dpp throughout the wing is not required for growth.

Further work will be needed to explain how the Dpp signal regulates the growth of the wing. The answer to this question will contribute to a better understanding of the role of morphogens in regulating the size of human organs and how a failure to do so might cause developmental disorders.

the increase of Dpp signaling levels is genetically blocked - were shown to grow at rates comparable to those of *wild-type* cells (*Schwank et al., 2012*). The alternative 'growth equalisation model' suggests that Dpp controls growth of the central region of the wing by repressing Brinker (*Schwank et al., 2011*). The recent use of membrane-tethered-GFP nanobodies against Dpp-GFP to modulate Dpp spreading proposes that Dpp emanating from the AP boundary is mainly required for the growth of the center of the wing disc, the region with the highest levels of Dpp activity (*Harmansa et al., 2015*). A recent report questions the temporal and spatial requirement of Dpp expression and activity in promoting wing growth. CRISPR-Cas9-mediated genome editing of the *dpp* locus combined with the FLP-FRT recombination system to temporally control the removal of Dpp from its endogenous stripe domain has come to the conclusion that Dpp emanating from the AP boundary is not required for wing growth and that the growth-promoting role of Dpp is restricted to the early stages of wing development (*Akiyama and Gibson, 2015*).

In order to carefully characterize the spatial and temporal requirement of Dpp in promoting proliferative growth of the different regions of the wing disc, here we combine the use of the conventional Gal4/UAS system and a temperature-sensitive Gal80 molecule together with a collection of RNAi hairpins to deplete *dpp* expression in the developing wing. We have carried out tissue size measurements in developing primordia and adult wings, FRT-mediated mitotic recombination clones and Gal4-based cell lineage analysis to present evidence that Dpp emanating from the AP compartment boundary is required throughout development to promote growth of the developing wing appendage by regulating cell number and tissue size. We also present experimental evidence supporting the proposal that Dpp promotes proliferative growth simply by maintaining Brinker levels below a growth-repressing threshold (*Schwank et al., 2008*), as nearly *wild-type* sized adult wings can be obtained by co-depleting Dpp and Brinker. Remarkably, our results indicate that Dpp regulates growth and proliferation rates equally in central and lateral regions of the developing wing appendage and that reduced levels of Dpp has an impact in both width and length of the wing.

Altogether, our results together with previous observations on the impact of Dpp spreading on wing growth (*Ferreira and Milán, 2015*) indicate that regional differences in Dpp signaling activity do not have a direct impact on growth rates, that graded activity of Dpp is not an absolute requirement for wing growth, and that the range of Dpp spreading emanating from the AP compartment boundary can regulate the final size of the wing appendage.

## Results

### Independent RNAi hairpins to deplete Dpp in the developing wing

We first drove expression of five RNAi hairpins, one long and four short, targeting various regions of the first *dpp* coding exon (*Perkins et al., 2015*) under the control of the *nubbin-gal4* (*nub-gal4*) driver. We selected this driver as its expression begins at the time the wing is being specified, namely in late second instar wing primordia (*Ng et al., 1995*, *1996*), and it is restricted to those cells that will give rise to the adult wing blade and the distal hinge (*Zirin and Mann, 2007*). Larvae were raised at 25°C and the resulting adult wings were analysed. Control individuals expressing an RNAi hairpin targeting GFP and containing the *nub-gal4* driver were raised in parallel in this and subsequent experiments. All individuals expressing the different *dpp-RNAi* hairpins produced rudimentary wings resembling those caused by the classical *dpp-disk* alleles in which the enhancers driving *dpp* expression to the imaginal tissues are removed [*Figure 1A,A'*; (*Masucci et al., 1990*)]. We observed that *dpp-RNAi*-expressing wings maintained largely well-formed and grown distal hinge structures (eg. alula and costa, *Figure 1A*), as occurs in *dpp-disk* wings (*Zecca et al., 1995*). In order to address the temporal requirement of Dpp activity, we next used the temperature-sensitive Gal80 molecule, which represses Gal4 transcriptional activity at low temperatures (18°C, [*McGuire et al., 2004*]). Larvae were raised at three temperatures (18°C, 25°C and 29°C) during larval and pupal development. Individuals raised at 18°C showed no overt wing phenotype (*Figure 1B,B'*), while those raised at 25°C showed only a mild growth phenotype (*Figure 1C,C'*). This observation reinforces the robust repression of Gal4 transcriptional activity by Gal80ts at low temperatures. All individuals raised at 29°C and expressing the different *dpp-RNAi* hairpins produced rudimentary wings (*Figure 1D,D'*). Based on the phenotypes at 25°C, we chose the strongest *dpp-RNAi* hairpin (BL 36779) for further experiments. We next characterised the effects of *dpp* depletion on the size of the developing wing appendage and monitored Dpp expression and activity levels. The developing appendage, labelled by the expression of a Nubbin (Nub) antibody, was drastically reduced in size when raised at 29°C (*Figure 1E*). We also used antibodies to Wingless (Wg) and Patched (Ptc) to label the dorsal-ventral (DV) and AP compartment boundaries and analyse the impact on the size of each compartment. The expression of Wg in two concentric rings (the outer, OR, and inner, IR, rings) in the proximal region of the wing disc helped us to delimit the wing pouch (the region that will give rise to the adult wing blade, inside the IR) from the surrounding wing hinge (between IR and OR, *Figure 1E*). In *dpp*-depleted wing primordia, the sizes of the D and V compartments were reduced in a proportional manner, which is consistent with the symmetric reduction in size of the D and V surfaces of the adult wing (*Figure 1D*). We observed that the size of the A compartment was reduced to a larger extent than the size of the P compartment (*Figure 1E*). The size of the wing hinge (the distance between the IR and OR, *Figure 1E*, bottom panels) was largely unaffected. As expected, *dpp* mRNA levels and Dpp activity levels, visualised with an antibody against the phosphorylated form of the Dpp transducer MAD (p-MAD) and by the expression of the Dpp target genes *spalt* and *optomotor-blind* [*omb*, (*de Celis et al., 1996*; *Lecuit et al., 1996*; *Nellen et al., 1996*)], decreased in the developing wing appendage (*Figure 1F–I*). Brinker (Brk), a transcription factor whose expression is repressed by the activity of Dpp and restricted to the lateral sides of the wing disc (*Campbell and Tomlinson, 1999*; *Jaźwińska et al., 1999*; *Minami et al., 1999*), was de-repressed in all wing pouch cells (*Figure 1G–I*).

### Dpp is continuously required to induce growth of the adult wing blade

We next performed temperature-shift experiments (from 18°C to 29°C) to initiate depletion of *dpp* at different times of larval and pupal development and to characterise the impact on the size of the resulting adult wings. We started our temperature shifts in the mid-second instar stage, as the wing appendage is specified and *nubbin* expression is initiated in late second instar wing discs [grey arrow

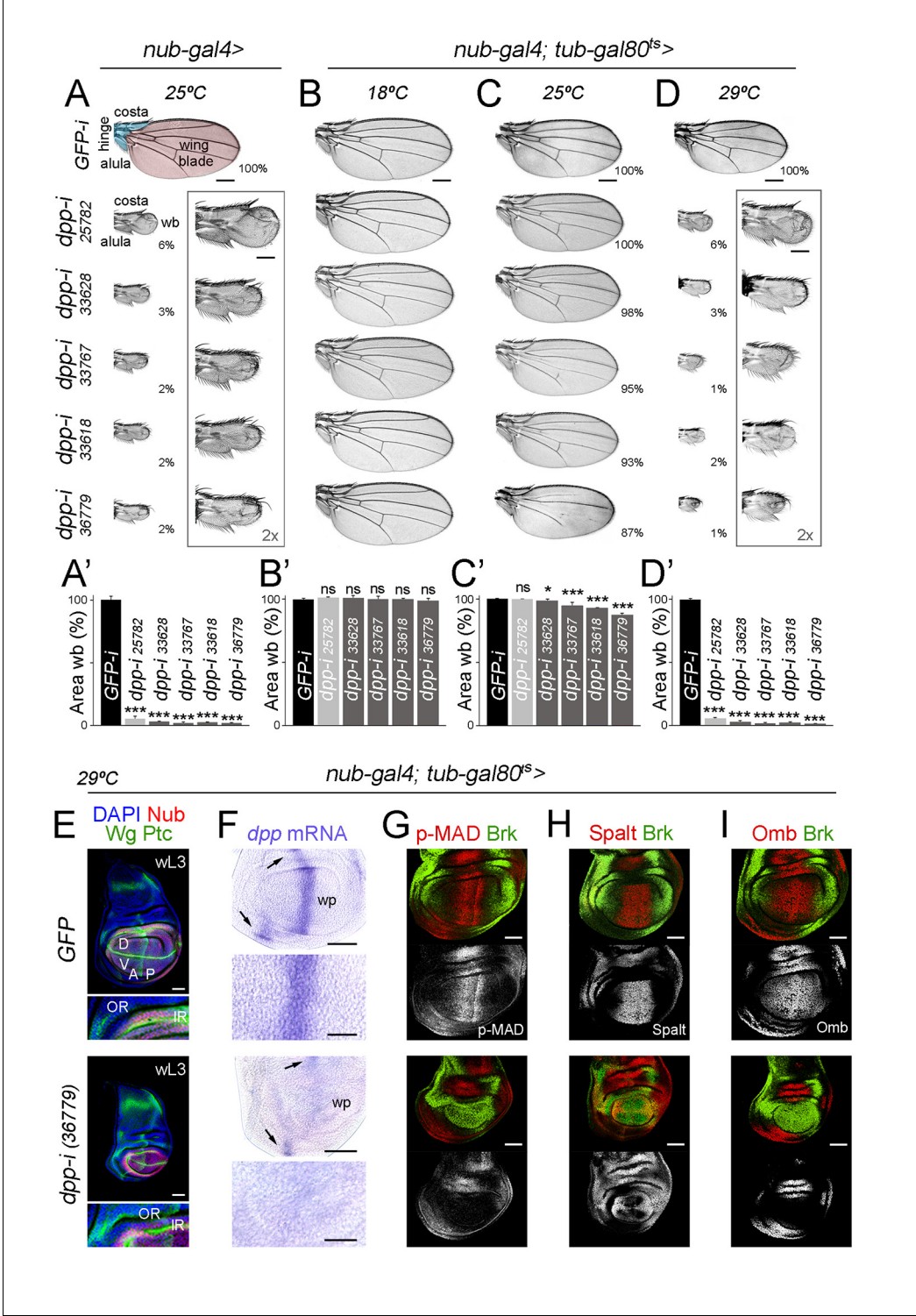

**Figure 1.** Independent RNAi hairpins to induce temporally controlled depletion of Dpp. (A–D) Cuticle preparations of male adult wings expressing the indicated RNAi hairpins under the control of *nub-gal4* and grown at the indicated temperatures. In B-D, flies carry the *tub-gal80ts* transgene. High magnification of rudimentary wings are shown in A and D, and percentages of wing size with respect to control *GFP-RNAi* expressing wings are indicated in A, C, and D. Wing blade (wb, pink) and hinge structures (alula and costa, blue) are shaded in A. Scale bars in A-D, 300 μm. Scale bars in the squared wings in A, D, 150 μm. (A′–D′) Histograms plotting tissue size of the wing blade with the indicated genotypes normalized as a percent of the control wings. Error bars show standard deviation. Number of wings per genotype and temperature >15. ns, not significant; ***p<0.001, *p<0.05. (E–I) Late

*Figure 1 continued on next page*

Figure 1 continued

third instar wing discs of the indicated genotypes, grown at 29°C, and stained for Wg and Ptc (E, green), DAPI (E, blue), Nub (E, red), *dpp* mRNA (purple, **F**), p-MAD (G, red), Spalt (H, red), Omb (I, red) and Brk (G-I, green). Scale bars, 50 µm (**E–I**) or 25 µm (higher magnifications in E, F). Higher magnifications of the dorsal hinge region are shown below each wing disc in panel E. In E, inner (IR) and outer (OR) rings of Wg, and dorsal (**D**), ventral (**V**), anterior (**A**) and posterior (**P**) compartments are marked. Note that the width of the hinge is largely unaffected by Dpp depletion. In F, *dpp* mRNA levels are reduced in the wing pouch (wp) when compared to the hinge region (black arrows). Higher magnifications of the wing pouch are shown below each wing disc in panel F.

The following source data is available for figure 1:

**Source data 1.** Summary of tissue size quantifications.

in *Figure 2A*, (*Ng et al., 1995*, *1996*)]. Depletion in pupal stages, when Dpp is expressed along the

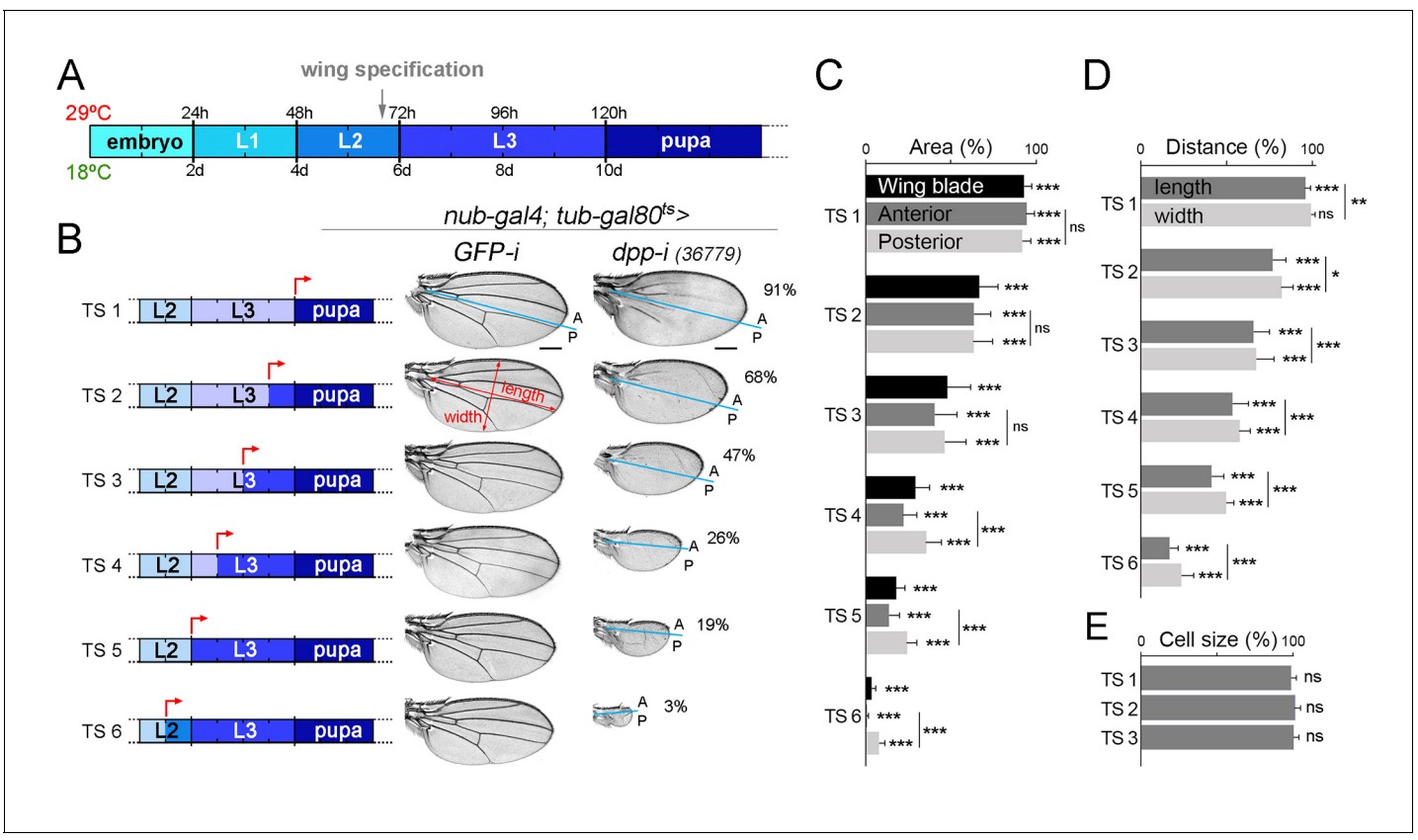

**Figure 2.** Dpp is continuously required for growth of the wing blade. (**A**) Cartoon depicting developmental timing in hours (h) and days (d) at 29°C and 18°C, respectively. Grey arrow marks the developmental timing at which the wing is specified. L1-L3, larval stages. (**B**) A series of cuticle preparations of male adult wings carrying the *tub-gal80ts* transgene and the *nub-gal4* driver and shifted from 18°C to 29°C at the developmental time points (red arrows) indicated in the corresponding cartoons to initiate expression of *GFP-* or *dpp-RNAi* hairpins until adulthood. The percentages of wing size with respect to control *GFP-RNAi* expressing wings subjected to the same temperature shifts are indicated. Anterior, A, and posterior, P, compartments are marked by blue lines based on the characteristic anterior-posterior pattern of bristles at the wing margin. Scale bars, 300 µm. (**C–E**) Histograms plotting tissue size (**C**), proportions (width and length, **D**), and cell size (**E**) of adult wings carrying the *tub-gal80ts* and the *UAS-dpp-RNAi* transgenes and the *nub-gal4* driver, shifted from 18°C to 29°C at the developmental time points TS1-TS6 indicated in the cartoons in B and normalized as a percent of the *GFP-RNAi* expressing control wings. Error bars show standard deviation. Number of wings per temperature >15. ***p<0.001; ns, not significant.

The following source data is available for figure 2:

**Source data 1.** Summary of tissue size and width and length quantifications.

presumptive vein regions (*de Celis, 1997*; *Yu et al., 1996*), had a mild impact on final wing size (*Figure 2B,C*). The earlier the initiation of Gal4-dependent expression of the *dpp-RNAi* hairpin in larval wing primordia, the stronger the effects on wing size (*Figure 2B,C*). Remarkably, the effects on tissue size increased in a gradual manner. As expected, cell size was largely unaffected in *dpp*-depleted wings (*Figure 2E*). In all cases, we observed that *dpp* depletion gave rise to a reduction in both the width and length of the wing resulting in smaller but largely well-proportioned adult structures (*Figure 2D*). The distal hinge structures were well formed and grown. At longer exposures, the size of the A compartment was reduced to a larger extent than the size of the P compartment (*Figure 2C*). All together, these observations indicate that *dpp* expression in wing cells is continuously required since the wing appendage is being specified and during the whole third instar stage to promote proliferative growth of the wing blade.

## Dpp emanating from the AP boundary is continuously required for growth of the wing pouch

CRISPR-Cas9-mediated genome editing of the *dpp* locus has enabled the conditional removal of the *dpp* gene from its endogenous stripe domain and contributed to the proposal that the Dpp morphogen gradient emanating from the AP compartment boundary is not continuously required to induce wing growth (*Akiyama and Gibson, 2015*). In order to revisit this proposal, we used the *dpp^disk^-gal4 (dpp-gal4)* driver to induce targeted expression of the *dpp-RNAi hairpin along the compartment boundary. This Gal4 driver contains the 4 kb DNA fragment within the 3'disk region called the blk* enhancer fragment, which is expressed in all *dpp*-producing cells in imaginal tissues (*Masucci et al., 1990*) and known to rescue *dpp-disk* alleles when driving Dpp expression (*Staehling-Hampton et al., 1994*). Once again, we used the *tub-gal80ts* transgene to modulate *dpp* depletion over time. We first characterised the kinetics of Dpp depletion by shifting the larvae from 18°C to 29°C at different time points of the third instar stage and analysing *dpp* expression and Dpp activity levels in late third instar wing discs (*Figure 3* and *Figure 3—figure supplement 1*). We used wing discs grown at 18°C as controls and monitored *dpp* mRNA and p-MAD, Brk, Spalt and Omb protein levels. p-MAD levels and Brk repression in the central part of the wing pouch were the first to be altered upon induction of *dpp-RNAi* expression (*Figure 3*). After 12 hr of induction, p-MAD levels were visibly reduced, and Brk initiated expression in the central part of the wing pouch. Twenty-four hours of induction were sufficient to completely remove p-MAD levels and obtain robust expression of Brk in all wing pouch cells. Longer exposures (at least 36 hr) to *dpp-RNAi* expression were required to remove the expression of the Dpp target genes Spalt and Omb from wing pouch cells (*Figure 3*). The levels of *dpp* mRNA were already decreased after a period of 12–24 hr of induction but longer exposures (36–48 hr) were required to induce a more robust, though not complete, reduction of *dpp* mRNA (*Figure 3—figure supplement 1*). These results indicate that the wing pouch levels of Brk and p-MAD proteins are highly sensitive to reductions in Dpp, and that Spalt and Omb either require stronger or longer reduction in Dpp activity levels or they are highly stable proteins, which might contribute to the robustness of Dpp-mediated patterning of the wing.

Unfortunately, adult wings expressing *dpp-RNAi* under the control of *dpp-gal4* could not be recovered due to early pupal lethality. We thus analysed and quantified the effects on tissue size on larval wing primordia and monitored both the size of the whole wing disc and of the developing wing appendage (wing pouch). We performed two protocols to deplete Dpp levels at the AP boundary and quantified the effects on tissue size (*Figure 4A,B* and *Figure 5A*). We first analysed the effects on the size of late third instar wing discs upon induction of *dpp-RNAi* expression at the AP boundary during the last 12, 24, 36 and 48 hr of larval development. We stained the wing primordia with Wg and Ptc antibodies to label the DV and AP compartment boundaries, respectively. Control individuals expressing GFP and containing the *dpp-gal4* driver and the *tub-gal80ts* transgene were subjected to the same temperature shifts and quantified in parallel. In this experimental setup, the folds of the wing hinge were used as a proxy to delimit the wing pouch (depicted in *Figure 4A*). The effects on the size of the wing pouch and of the whole wing disc were already observed after 24 hr of *dpp-RNAi* expression and longer exposures gave rise to a gradual reduction in wing pouch and wing disc size (*Figure 4A–D*). The impact on the size of the wing pouch was stronger than on the size of the whole wing disc, thereby indicating that Dpp plays a major role in promoting the growth of the wing appendage and a minor role in the growth of the body wall (see below). The size of the A compartment was also reduced to a larger extent than the size of the P compartment (*Figure 4—*

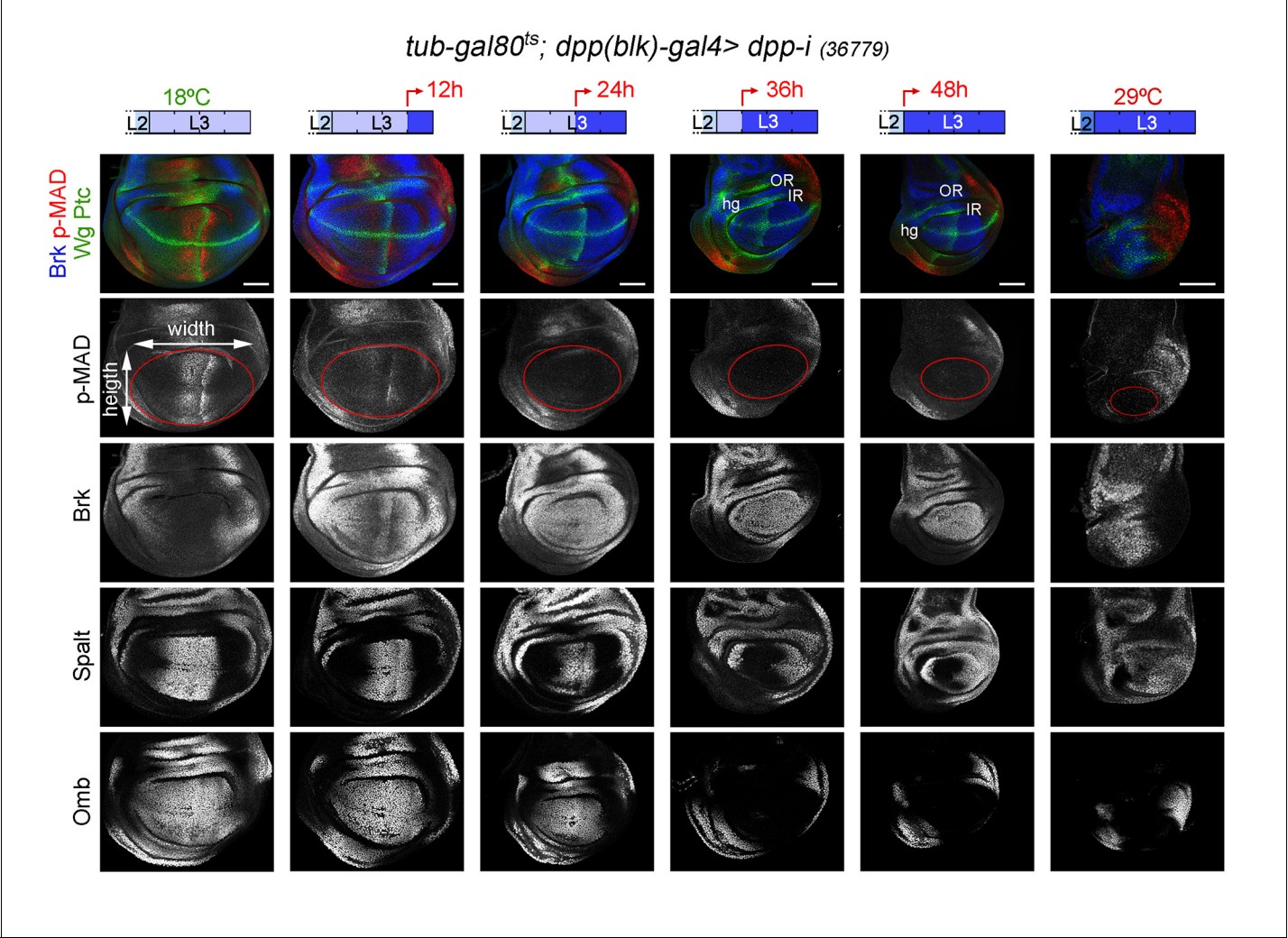

**Figure 3.** Changes in Dpp signaling and target gene expression upon temporal depletion of boundary Dpp. Late third instar wing discs of the indicated genotypes, raised at 18°C or 29°C throughout development (left or right panels), or shifted from 18°C to 29°C at the indicated developmental times (red arrows), and stained for Wg and Ptc (green), p-MAD (red or grey), Brk (blue or grey), Spalt (grey) and Omb (grey). A proxy of the wing pouch was marked by a red line. The inner (IR) and outer (OR) rings of Wg, and the hinge (hg) are marked in some wing discs. Note that the width of the hinge is largely unaffected by Dpp depletion. Scale bars, 50 µm. L2, L3, second and third instar.

The following figure supplement is available for figure 3:

**Figure supplement 1.** *dpp mRNA* expression upon temporally controlled expression of *dpp-RNAi* with the *dpp(blk)-gal4* driver.

*figure supplement 3*). Similar results were obtained with other *dpp-RNAi* lines (*Figure 4—figure supplement 3*). The overall proportions of the wing pouch (width versus length, *Figure 3* and *Figure 4A,B*) and the width of the wing hinge (the distance between the IR and the OR, *Figure 3*, *Figure 4B* and *Figure 4—figure supplement 1*) were largely unaffected by *dpp* depletion. Temporally controlled expression of *dpp-RNAi* with *ptc-gal4*, which is restricted to the AP compartment boundary, also caused a stronger reduction of the wing pouch than of the wing disc (*Figure 4—figure supplement 2*). In all cases, at 18°C, the size of the wing pouch and the wing disc was unaffected in *dpp-RNAi*-carrying discs.

The second approach used was to initiate *dpp-RNAi* expression at the AP boundary in early third instar wing discs and to quantify the impact on tissue size 12, 24, 36 and 48 hr thereafter (*Figure 5A*). Control individuals expressing GFP and containing the *dpp-gal4* driver and the *tub-gal80ts* transgene were subjected to the same temperature shifts and quantified in parallel.

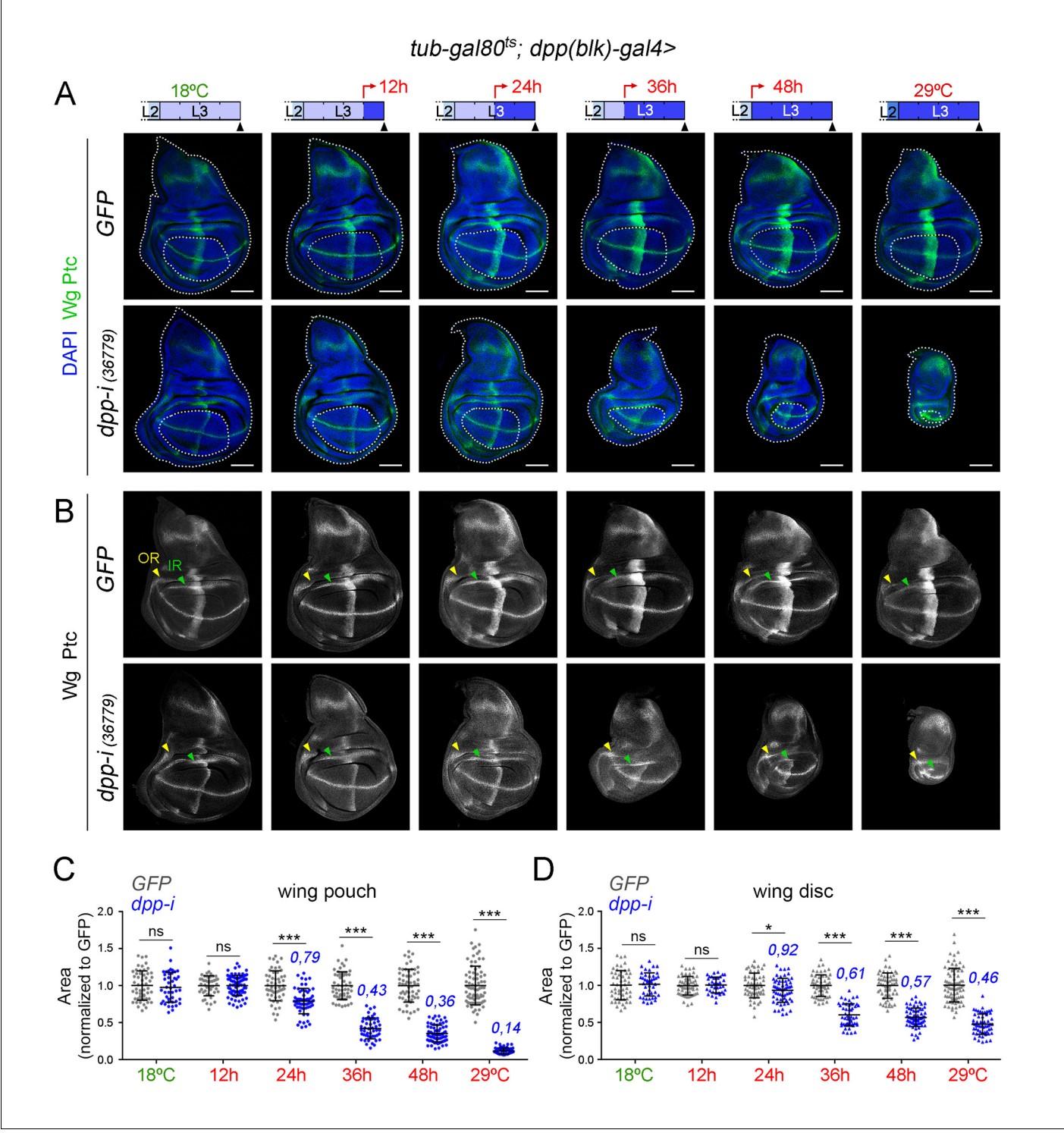

**Figure 4.** Effects on the size of the wing pouch and wing disc upon temporal depletion of boundary Dpp. (**A, B**) Late third instar wing discs of the indicated genotypes, raised at either 18°C or 29°C throughout development (left and right panels) or shifted from 18°C to 29°C at the indicated developmental times (red arrows). Discs were stained for Wg and Ptc (A, green or B, white), and DAPI (A, blue). In **A**, wing pouch and disc contours are marked by a dotted line. In **B**, the outer, OR, and inner, IR, rings of Wg are marked by arrowheads and used to delimit the hinge (between the rings). Note the width of the hinge is largely unaffected by Dpp depletion. Scale bars, 100 μm. (**C, D**) Scatter plots showing the size (normalized to GFP) of the wing pouch (**C**) and wing disc (**D**) of the indicated genotypes, raised at either 18°C or 29°C throughout development or shifted from 18°C to 29°C at the indicated developmental times shown in A. Average wing pouch or wing disc areas of *dpp-RNAi*-expressing individuals (normalized to GFP) are shown

*Figure 4 continued on next page*

*Figure 4 continued*

in blue. Error bars show standard deviation. Number of wing discs per experiment: n(GFP)=45–75; n(dpp-i)=40–70. \*\*\*p<0.001, \*p<0.05; ns, not significant.

The following source data and figure supplements are available for figure 4:

**Source data 1.** Summary of tissue size quantifications.

**Figure supplement 1.** Different effects of Dpp-depletion on the size of the wing pouch and hinge.

**Figure supplement 2.** Boundary Dpp is required for growth of the wing disc.

**Figure supplement 3.** Effects on the size of the wing pouch, wing disc and anterior and posterior compartments upon temporal depletion of boundary Dpp with different *dpp-RNAi* lines.

**Figure supplement 3—source data 1.** Summary of tissue size quantifications.

Expression of Wg at the inner ring (IR) of the wing hinge helped us to delimit the wing pouch in young and mature wing discs (*Figure 5A*). After 12 hr of induction, p-MAD was completely lost, and Brk initiated expression in the central part of the wing pouch (*Figure 5—figure supplement 1*). Twenty-four hours of induction were sufficient to obtain robust expression of Brk in all wing pouch cells. Again, the effects of temporally controlled boundary expression of *dpp-RNAi* on the size of the wing pouch were stronger than on the size of the wing disc (*Figure 5A–C*). The wing pouch of control wing discs expressing GFP showed an eleven-fold increase in size during the first half of the third instar but only a two-fold increase during the last 24 hr of larval development (*Figure 5B*). Interestingly, the impact of Dpp depletion on the size of the wing pouch and wing disc was still observed during this later developmental period (*Figure 5A–C*). These results reinforce the requirement of Dpp for wing pouch growth throughout the third instar larval period (*Figure 2*) and suggest that the minor impact on wing pouch size observed in the first protocol upon 12-h depletion of Dpp (*Figure 4A,B*) should be explained by insufficient reduction of *dpp* mRNA levels (*Figure 3—figure supplement 1*).

Dpp is a target of Hedgehog (Hh) coming from the P compartment (*Capdevila et al., 1994*; *Zecca et al., 1995*). In mature third instar wing discs, Dpp is expressed in a narrow stripe at the AP boundary. In young third instar discs, the Dpp stripe has a similar width but it proportionally occupies one third of the disc (*Weigmann and Cohen, 1999*). Thus, cells lose expression of Dpp when they are displaced out of range of the Hh signal by growth of the disc. It has recently been shown that transient exposure of cells to short hairpins can reduce gene function in their descendants, due to the high stability of these hairpins (*Bosch et al., 2016*). In order to rule out the possibility that the effects on tissue size are caused by targeted depletion of *dpp* in a broader domain than the one expressing *dpp* at the time of visualisation, we used the G-TRACE technique to label the domain that has been exposed to *dpp-RNAi* expression during the induction process (*Evans et al., 2009*). In wing discs carrying the *tub-gal80ts* transgene and expressing a *dpp-RNAi* hairpin in the *dpp-gal4* domain during 12, 24, 36 and 48 hr, the presence of an *UAS-FLP* transgene together with an FRT-mediated FLP-out cassette driving EGFP expression (*ubi-FRT-stop-FRT-EGFP*), and an *UAS-RFP* construct, allowed us to irreversibly label, in green, all cells born in the *dpp-gal4* domain during the induction process, and to compare this domain with the one expressing the transgenes, in red, at the time of visualisation (*Figure 6A,B*). Control individuals not expressing a *dpp-RNAi* hairpin were subjected to the same protocol and analysed in parallel. As expected, the longer the induction, the broader the domain labelled by EGFP, thereby reinforcing the notion that cells lose Dpp expression when they are displaced out of range of the Hh signal by growth of the disc (*Figure 6A,B*). In the case of induction periods of 24 hr or 36 hr, this domain occupied a small fraction of the central region of the wing primordium, and the non-autonomous effects on tissue size were significant, visualised by the impact on the size of the P compartment and on the size of the A compartment not labelled by EGFP (*Figure 6B,C*). Although it is still possible that some low level of Dpp made outside the stripe might be knocked down in our experiments, our G-TRACE and temperature shifts

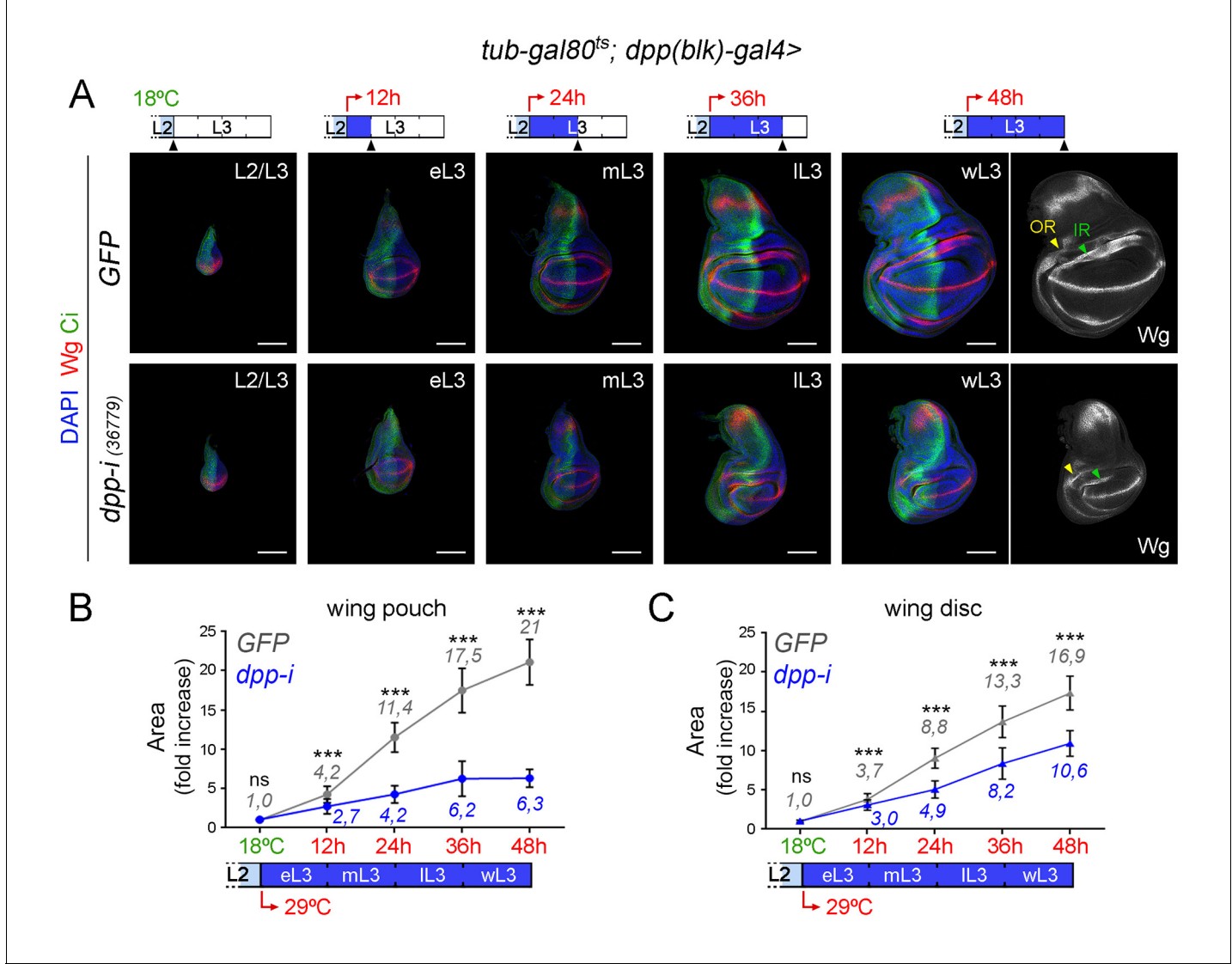

**Figure 5.** Effects on growth rates in the wing pouch and wing disc upon boundary depletion of Dpp. (A) Larval wing discs of the indicated genotypes shifted from 18°C to 29°C at the L2/L3 transition (red arrows) and dissected 0, 12, 24, 36 and 48 hr thereafter. Discs stained for Wg (red) and Ci (green), and DAPI (blue). Scale bars, 100 µm. L2, L3, eL3, mL3, lL3, wL3: second, third, early-third, mid-third, late-third and wandering-third larval stages. In the right panel, the outer, OR, and inner, IR, rings of Wg are marked by arrowheads and used to delimit the hinge (between the rings). Note the width of the hinge is affected by Dpp depletion to a lesser extent than the pouch. (B, C) Fold change increase in the size of the wing pouch and wing disc (with respect to the one at the L2/L3 transition) of individuals expressing *dpp-RNAi* (blue) or GFP (grey) shifted from 18°C to 29°C at the L2/L3 transition (red arrows). The average fold change increases in the area of the wing pouch (B) and wing disc (C) with respect to the values at the beginning of the temperature shift at the L2/L3 transition are indicated. Error bars represent standard deviation. Number of wing discs per developmental point: n (GFP) = 25–45; n (dpp-i) = 30–40. ***p<0.001; ns, not significant.

The following source data and figure supplement are available for figure 5:

**Source data 1.** Summary of tissue size quantifications.

**Figure supplement 1.** Changes in Dpp signalling in early wing discs upon temporal depletion of boundary Dpp.

results (*Figures 4*, *5* and *6*) support the proposal that Dpp emanating from the AP compartment boundary is the one required to induce wing growth since the wing is being specified. Our experimental observation that incomplete reductions in the levels of *dpp* mRNA at the AP compartment boundary (*Figure 3—figure supplement 1*) can have a clear impact on the size of the wing pouch

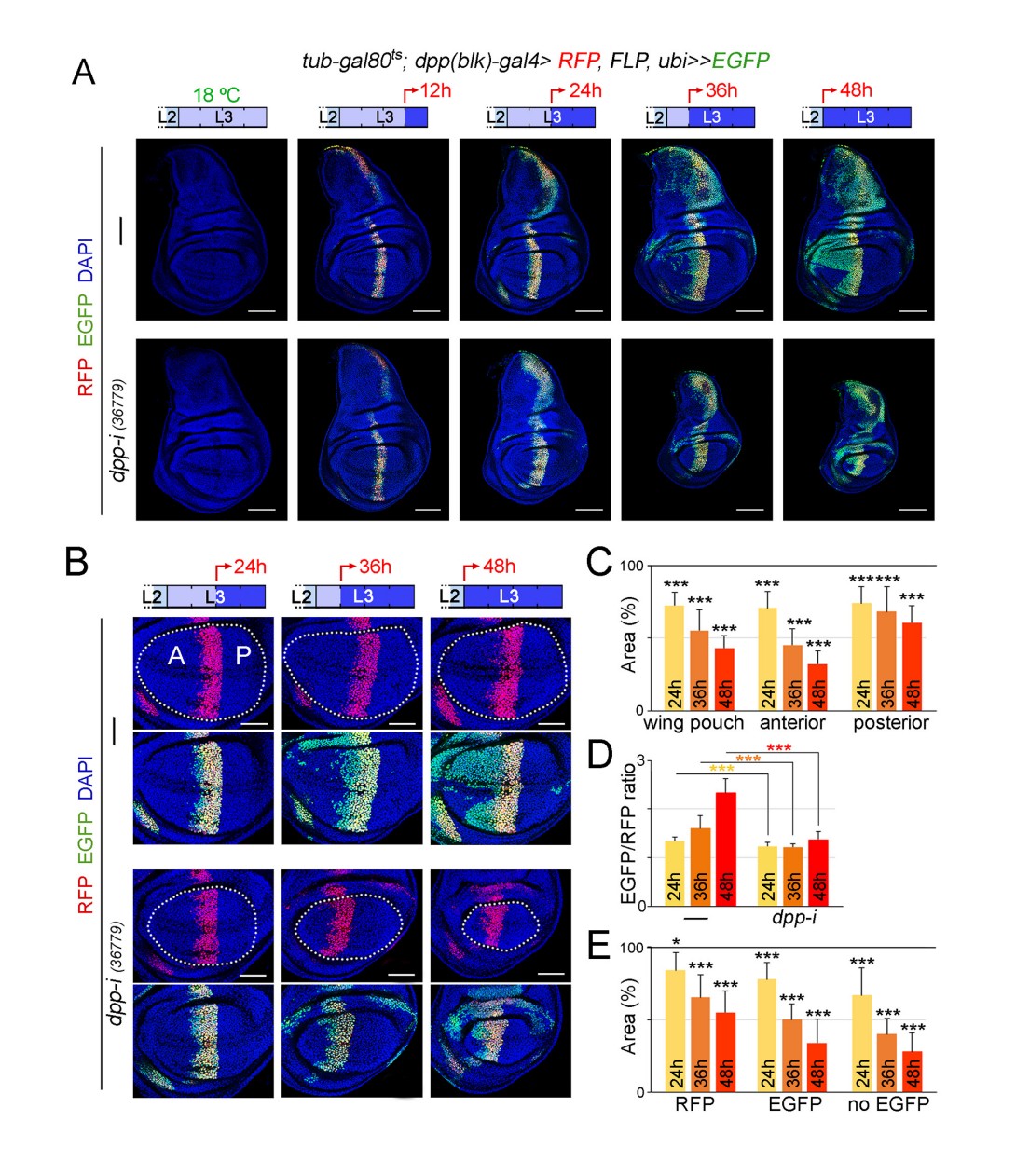

**Figure 6.** Cell lineage analysis of *dpp-RNAi* expressing cells and effects on growth rates upon temporal depletion of boundary Dpp. (A) G-TRACE-mediated cell lineage analysis to irreversibly label all cells born in the *dpp-gal4* expressing domain during the last 12, 24, 36 and 48 hr of larval development. Larvae were shifted from 18°C to 29°C at the developmental times indicated in the cartoons (red arrows) to express *dpp-RNAi*, and late third instar wing discs were stained for RFP (red), EGFP (green) and DAPI (blue). As a proof of concept, larval wing discs were raised at 18°C (left panel). Scale bars, 100 μm. L2, L3, second and third larval stages. (B) G-TRACE to label all cells born in the *dpp-gal4*-expressing domain during the last 24–48 hr of larval development. Larvae were shifted from 18°C to 29°C at the indicated developmental times (red arrows) and late third instar wing discs were stained for RFP (red), EGFP (green) and DAPI (blue). Wing pouch contours are marked by a dotted line. Scale bars, 50 μm. L2, L3, second and third instar. A, anterior; P, posterior compartments. (C–E) Histograms plotting the size of the indicated regions (normalised to those of control discs not expressing *dpp-RNAi*, (C, E) or the size ratio between the EGFP- and RFP-expressing domains (D) in late third instar wing pouches (inside dotted line) expressing *dpp-RNAi* during the last 24–48 hr of larval development. Number of discs per genotype and induction time = 20–35. Error bars represent standard deviation, and *p<0.05; ***p<0.001.

The following source data is available for figure 6:

**Source data 1.** Summary of tissue size and EGFP/RFP ratio quantifications.

(*Figure 4*) is in conflict with the previously proposed role of low levels of non-boundary Dpp as the ones promoting wing growth (*Akiyama and Gibson, 2015*), and it strongly suggests that wing growth requires high levels of boundary Dpp.

## Dpp is equally required for the proliferative growth of both medial and lateral regions of the developing wing appendage

We made use of the G-TRACE results to characterise the requirement of Dpp emanating from the AP boundary on the growth of the different regions of the wing pouch. We analysed wing discs subjected to *dpp* depletion for 24, 36 and 48 hr. The impact on the size of the A compartment of the wing pouch was stronger than on the size of the P compartment (*Figure 6B,C*). The ratio between the EGFP and RFP expression domains, a proxy of the expansion of the medial region, was reduced in *dpp*-depleted wing discs when compared to controls and did not increase over time (*Figure 6D*). Within the A compartment of the wing pouch, we observed that the size of the EGFP-expressing domain was reduced to a similar extent than the size of the domain not expressing EGFP (*Figure 6E*). These results suggest that Dpp is equally required for the growth of both medial and lateral regions of the developing wing appendage.

In order to further address the impact of Dpp depletion on the growth rates of medial and lateral regions of the developing wing appendage, we induced neutral clones of cells in early-second instar, initiated *dpp-RNAi* expression in early-third instar wing discs and examined the size of these clones 42 hr later in mature wing discs. Clone size (in arbitrary units) and number of cells per clone were measured in *hs-FLP, RFP, FRT19A/FRT19A; tub-gal80ts; dpp-gal4/UAS-dpp-RNAi* wing discs, and these two measurements were compared to those of control clones induced in *hs-FLP, RFP, FRT19A/FRT19A; tub-gal80ts; dpp-gal4/UAS-EGFP-RNAi* wing discs and grown in parallel. In *dpp*-depleted wing discs, the size of the clones and the number of cells per clone in the wing pouch was, as expected, significantly smaller than the size of clones quantified in the wing pouch of *EGFP-RNAi*-expressing discs (*Figure 7A,B*). The size reduction of posterior clones was slightly less than the one observed in anterior clones (*Figure 7B*). Both medial and lateral clones were smaller to a similar extent, thus reinforcing the notion that Dpp is equally required for the growth of both medial and lateral regions of the developing wing pouch (*Figure 7C*). We observed that the elongated shape of the clones along the proximal-distal axis of the wing pouch was largely unaffected by depletion of *dpp* (*Figure 7A*). The size of clones of cells in the body wall and hinge regions of *dpp*-depleted wing discs was reduced to a smaller extent than those located in the wing pouch (*Figure 7D* and *Figure 7—figure supplement 1*). In order to further characterize the impact of reduced Dpp on the growth of medial and lateral regions of the wing, we also examined the size distribution of clones in pupal wings. In this case, we generated neutral clones of cells and initiated *dpp-RNAi* expression in mid-third instar wing discs of *hsFLP; ubi-FRT-stop-FRT-GFP/nub-gal4; tub-gal80ts /UAS-dpp-RNAi* individuals, and the number of cells per clone was compared to that of control clones examined in *hsFLP; ubi-FRT-stop-FRT-GFP/nub-gal4; tub-gal80ts/UAS-EGFP-RNAi* pupal wings. Interestingly, Dpp depletion caused a similar reduction in the size distribution of clones located in medial and lateral regions of the wing (*Figure 7E,F,F'*). We also observed that the size distribution of clones was nearly identical in medial and lateral regions of both control and *dpp*-depleted pupal wings (*Figure 7E,F,F'*). All together, these results indicate that Dpp emanating from the AP boundary is equally required to promote proliferative growth of medial and lateral regions of the developing wing.

## The Dpp gradient is not absolutely required for wing growth

Our experimental observations showing that temporally controlled depletion of Dpp gives rise to smaller but largely well-proportioned adult wings (*Figure 2B,D*) and that Dpp is equally required to promote proliferative growth of medial and lateral regions of the wing primordium (*Figure 7E,F,F'*) suggest that regional differences in Dpp signaling activity do not have a direct impact on growth rates. We next tested whether our results are consistent with the model proposed in *Schwank et al. (2008)* that indicates that Dpp regulates wing growth by simply counteracting the growth-repressive activity of Brinker. In this work, authors focused on the developing wing imaginal disc to present evidence that a hypomorphic allele of *brinker* was able to largely rescue the tissue size defects caused by a hetero-allelic and hypomorphic combination of *dpp* alleles. We took advantage of our

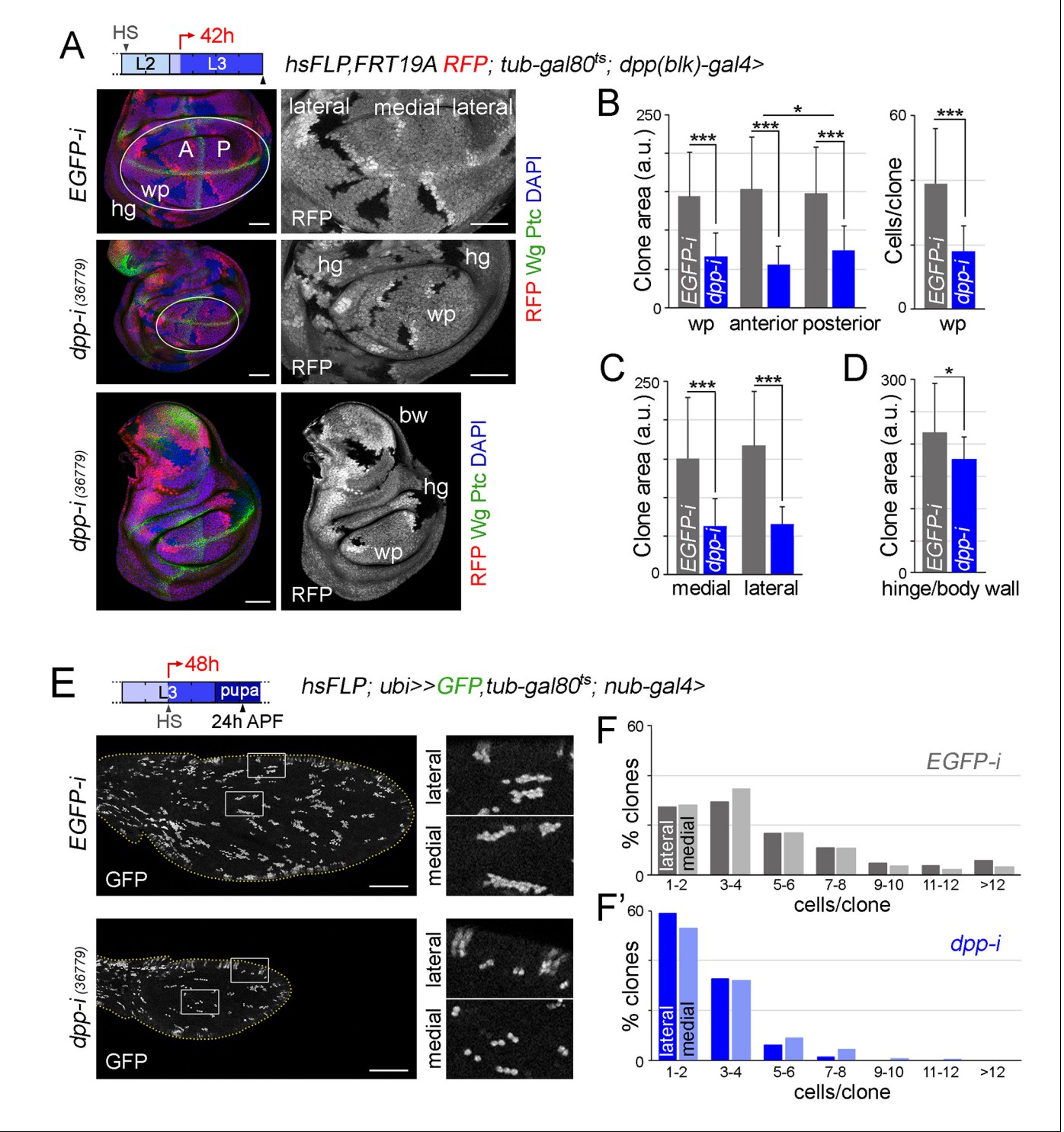

**Figure 7.** Boundary Dpp regulates growth and proliferation rates equally in medial and lateral regions of the developing wing. (A, E) Late third instar wing discs (A) or pupal wings (E) expressing the indicated transgenes for 42 hr (A) or 48 hr (E), and bearing neutral clones (labelled by the absence of RFP in red or white in A, or by the expression of GFP in white in E) induced in early second (A) or mid-third (E) instars. In A, discs were stained for Wg and Ptc (green) and DAPI (blue). In E, high magnification of the regions in the white boxes are shown on the right panels. Scale bars, 50 μm (A) or 100 μm (E). L2, L3, second and third instar. A, anterior; P, posterior compartments; wp, wing pouch; hg, hinge; APF, after puparium formation. (B–D) Histograms plotting clone size (in arbitrary units, B–D) or cells per clone (B, right) in the indicated regions of late third instar wing discs expressing the indicated transgenes. Clones were induced in early second instar, transgenes were expressed for 42 hr, and wing discs were dissected in late third instar. Number of clones: n (pouch) = 70–85, n (hinge/body wall)=35–40. Error bars represent standard deviation, and *p<0.05; ***p<0.001. (F, F')

*Figure 7 continued on next page*

*Figure 7 continued*

Distribution of cells per clone in the indicated regions of pupal wings expressing the indicated transgenes. Clones were induced in mid-third instar, transgenes were expressed for 48 hr, and pupal wings were dissected 24 hr APF. Number of clones in F: n (lateral) = 427, n (medial) = 855; number of clones in F': n (lateral) = 337, n (medial) = 404.

The following source data and figure supplement are available for figure 7:

**Source data 1.** Summary of clone size and number of cells per clone quantifications.

**Figure supplement 1.** Different effects of Dpp-depletion on the growth rates of wing pouch and hinge cells.

experimental setup based on expression of *dpp-RNAi* hairpins under the control of the *nub-gal4* driver to co-express one or two independent *brk-RNAi* hairpins and analyse the impact on the size of the resulting adult wings. Consistent with the results obtained by *Schwank et al. (2008)*, co-depletion of *brk* and *dpp* gave rise to nearly *wild type*-sized wing pouch primordia (*Figure 8A*). As expected, p-MAD levels were not rescued in these experimental conditions, and Spalt and Omb expression was restored, reinforcing the Dpp-Brinker double repression mechanism involved in Dpp

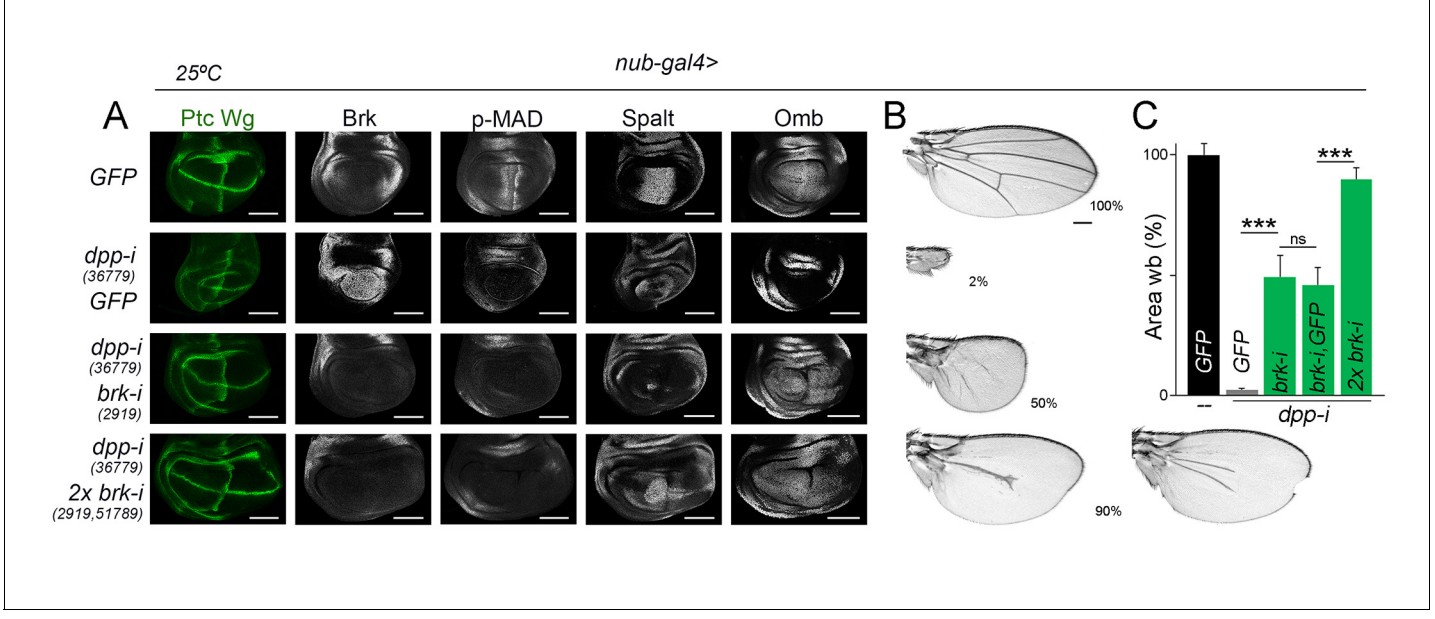

**Figure 8.** Wing growth in the absence of graded activity of Dpp. (**A**) Late third instar wing discs of the indicated genotypes, and stained for Wg and Ptc (green), and Brk, p-MAD, Spalt and Omb (grey). Scale bars, 100 μm. (**B**) A series of cuticle preparations of male adult wings of the indicated genotypes. Percentages of wing size with respect to control *GFP*-expressing wings are indicated. Scale bars, 300 μm. (**C**) Histograms plotting tissue size of adult wing blades (wb) carrying the indicated transgenes and the *nub-gal4* driver, and normalized as a percent of the GFP-expressing control wings. Error bars show standard deviation. Number of wings per temperature = 17–30. \*\*\*p<0.001; ns, not significant. Individuals were grown at 25°C.

The following source data and figure supplements are available for figure 8:

**Source data 1.** Summary of tissue size quantifications.

**Figure supplement 1.** Wing growth in the absence of graded activity of Dpp.

**Figure supplement 1—source data 1.** Summary of tissue size quantifications.

**Figure supplement 2.** Dpp spreading and wing growth.

**Figure supplement 2—source data 1.** Summary of tissue size quantifications.

target gene expression (*Affolter and Basler, 2007*; *Restrepo et al., 2014*). The use of two *brk-RNAi* lines at the same time was able to rescue to a larger extent the expression levels of Spalt (*Figure 8A*). We observed, although, that Spalt and Omb were expressed at lower levels than in *wild-type* primordia and that their expression profiles did not show their characteristic graded expression along the anterior-posterior axis (*Figure 8A*). Most interestingly, size but not patterning of the resulting adult wings was rescued by co-depletion of *dpp* and *brk*, and this rescue was almost complete by the use of two *brk-RNAi* lines at the same time (*Figure 8B,C*). Similar results on Dpp signaling and tissue size were obtained with the *rotund-gal4* driver, whose expression is also restricted to the developing wing blade (*Figure 8—figure supplement 1*). These observations indicate that graded activity of Dpp is required for wing patterning but not for wing growth, and reinforce the proposal that Dpp promotes growth of the wing blade mostly by reducing Brinker levels. We noticed that the shape of the wing pouch of *dpp* and *brk* co-depleted wing discs was elongated along the anterior-posterior axis (*Figure 8A* and *Figure 8—figure supplement 1*). This elongation was not observed in the resulting adult wings (*Figure 8B* and *Figure 8—figure supplement 1*), raising the possibility that cell re-arrangements during metamorphosis might contribute to the correction of the final wing shape.

## Discussion

Here, we have used a collection of *dpp-RNAi* hairpins and the GAL4/UAS system combined with the temperature-sensitive Gal80 molecule to control the depletion of *dpp* expression in time and space and characterize the spatial and temporal requirement of Dpp in promoting the growth of the different regions of the wing primordium: the wing blade, the wing hinge and the body wall. We present evidence that the impact of Dpp depletion on the growth rates and size of the hinge and body wall regions was mild. Our data indicate that Dpp emanating from the AP compartment boundary is absolutely required throughout development to promote the growth of the developing wing blade by regulating cell proliferation and growth rates. Dpp depletion gave rise to a reduction in the size of both the anterior-posterior and proximal-distal axes of the wing. We observed that growth rates in *wild-type* primordia were similar in those regions with the highest (medial) and lowest (lateral) levels of Dpp activity, and that Dpp depletion caused an identical impact on the growth rates of these two regions. These results argue against a direct relationship between Dpp signalling activity levels and the rates of proliferative growth within the developing wing blade.

The fact that co-depletion of Dpp and Brinker in developing wing blade cells gives rise to nearly *wild type*-sized adult wings indicate that graded activity of Dpp is not an absolute requirement for growth and that Dpp promotes proliferative growth by maintaining Brk expression levels below a growth-repressive threshold (*Schwank et al., 2008*). This proposal is consistent with the observation that wing blade cells unable to transduce the Dpp signal and mutant for *brk* proliferate at the same rate as *wild-type* cells (*Schwank et al., 2012*) and that loss of Brinker in clones of cells induces overgrowths mainly in the wing hinge region (*Campbell and Tomlinson, 1999*; *Jaźwińska et al., 1999*; *Martín et al., 2004*; *Minami et al., 1999*). These results question the classical 'steepness' model that proposes that juxtaposition of cells sensing disparate levels of the morphogen promotes proliferative growth (*Lawrence and Struhl, 1996*; *Rogulja and Irvine, 2005*) and the more recent 'temporal rule' model that postulates that cells divide when Dpp signaling levels have increased by 50% (*Wartlick et al., 2011*).

How is then the size of the wing controlled by Dpp? Of remarkable interest is the capacity of overexpression of Dally, a proteoglycan that contributes to Dpp stability (*Akiyama et al., 2008*), to induce the expansion and flattening of the Dpp gradient and to cause an increase in wing size [(*Ferreira and Milán, 2015*), *Figure 8—figure supplement 2*]. The observation that wing primordia overgrow even though the Dpp gradient is flattened also questions the classical 'steepness' model. A reduction in Dally expression levels induces smaller but well-proportioned wings in which the patterning elements are correctly located [(*Ferreira and Milán, 2015*), *Figure 8—figure supplement 2*]. These results support the proposal that the range of Dpp spreading emanating from the AP boundary can regulate, in an instructive manner, the final size of the developing wing appendage. We noticed that the effects of Dpp depletion on tissue size and growth rates were always stronger in the A compartment. Whether the differential response of P and A cells to Dpp depletion relies on the expression of the selector genes conferring compartment identity (*Tabata et al., 1995*) or on

the fact that Hedgehog is expressed in P cells but only sensed in A cells (*Domínguez et al., 1996*) remains to be elucidated.

The Cre-Lox and FLP-FRT recombination systems are widely used in developmental biology to induce conditional null alleles and address gene function. The recent use of an FRT-dependent conditional null allele of *dpp* came to the unexpected conclusion that Dpp emanating from the AP boundary is not required for wing growth and that the growth-promoting role of Dpp is restricted to the early stages of wing development (*Akiyama and Gibson, 2015*). Although authors conclude, in line of our results, that the Dpp gradient does not play an instructive role in wing growth, our results, based on the use of RNAi hairpins targeting *dpp*, do not validate their conclusions on the temporal and spatial requirement of Dpp expression and their impact on growth. It is interesting to note in this context that our work has demonstrated an important role of Dpp in promoting wing blade growth, and a minor role in promoting the growth of wing hinge and body wall. Unfortunately, Akiyama and Gibson based their conclusions only on the impact of Dpp removal on the size of the whole wing primordium and not specifically on the size of the developing wing blade. It is also certainly possible that RNAi-mediated depletion of gene activity might bypass the potential stability of the *dpp* mRNA, thus inducing a more efficient and rapid removal of Dpp activity than the FRT-dependent conditional null allele of *dpp*.

## Materials and methods

### Fly stocks

*The following strains were provided by the* Bloomington *Drosophila* Stock Center: *UAS-dpp^RNAi* (RRID:BDSC_25782, RRID:BDSC_33628, RRID:BDSC_33767, RRID:BDSC_33618, RRID:BDSC_36779), *UAS-brk^RNAi* (RRID:BDSC_51789), *UAS-GFP^RNAi* (RRID:BDSC_9331), *UAS-EGFP^RNAi* (RRID:BDSC_35782), *UAS-GFP* (RRID:BDSC_35786), *UAS-Red, UAS-FLP, ubi-FRT-stop-FRT-EGFP* (G-TRACE, RRID:BDSC_28282), *ubi-FRT-stop-FRT-GFP* (RRID:BDSC_32250), *nubbin-gal4* (RRID:BDSC_25754), and *UAS-dally^RNAi* (RRID:BDSC_33952), or by the Vienna *Drosophila* RNAi Center: *UAS-brk^RNAi* (# 2919), and *UAS-dally^RNAi* (#14136). *rotund-gal4* is described in *Colombani et al. (2012)*, *dpp^disk^-gal4* (*dpp-gal4 in the text*) in *Staehling-Hampton et al. (1994)*, and *UAS-dally* in *Ferreira and Milán (2015)*. Other stocks are described in Flybase.

### Immunohistochemistry

Mouse anti-Wg (1:50, 4D4, DSHB); mouse anti-Ptc (1:50, (Apa1, DSHB); rabbit anti-Tsh (1:600, gift from S. Cohen); rabbit anti-Nub (1:600, gift from X. Yang); rabbit anti-Spalt (1:500, gift from R. Barrio); rabbit anti-p-MAD (1:200) and guinea pig anti-Brk (1:1000; gift from G. Morata); rabbit anti-Omb (1:1000, gift from G. Pflugfelder); rat anti-Ci (1:10, 2A1, DSHB); and goat anti-Hth (1:50, sc-26187, Santa Cruz). Secondary antibodies Cy2, Cy3, Cy5 and Alexa 647 (1:400) were obtained from Jackson Immuno-research. A digoxigenin (DIG)-labelled antisense probe was transcribed by T3 RNA polymerase from an EcoRI digested full-length dpp cDNA clone RE20611 (gift from C. Estella), using the DIG RNA Labelling Kit (Roche) according to the manufacturer's instructions. In situ hybridisation was performed as in *Milán et al. (1996)*.

### Temporal and regional gene expression targeting (TARGET) with tub-gal80^ts

We used the Gal4/UAS system (*Brand and Perrimon, 1993*), combined with the thermo-sensitive version of Gal80 (Gal80ts, [*McGuire et al., 2004*]), a repressor of Gal4 protein activity, to precisely control, in time and space, the expression of *dpp-RNAi*. Adult flies carrying a Gal4 driver, the *tub-gal80ts* construct and the *UAS-dpp-RNAi* (experimental condition) were allowed to lay eggs in plates at 18°C over a period of 12 hr. Flies not carrying the *UAS-dpp-RNAi* transgene were also allowed to lay eggs in parallel (control condition). The progeny of both the experimental and control conditions was then raised at 18°C to maintain the Gal4/UAS system switched off and then transferred to 29°C for different periods during larval development to induce Gal4/UAS-dependent gene expression. Larvae were staged at the second to third larval stage transition to target *dpp-RNAi* expression for defined periods of time during third instar. Experimental conditions and control individuals were grown in parallel. In the case of targeted expression of *dpp-RNAi* in the *dpp-gal4* domain, G-TRACE

was performed as described in *Evans et al. (2009)* to trace the lineage of all the cells that have expressed *dpp-RNAi* during the induction period at 29°C. For this purpose, the following genotypes were used: (1) *UAS-Red, UAS-FLP, ubi-FRT-stop-FRT-EGFP/tub-gal80ts; dpp-gal4/+* and (2) *UAS-Red, UAS-FLP, ubi-FRT-stop-FRT-EGFP/tub-gal80ts; dpp-gal4/UAS-dpp-RNAi.*

## Quantification of tissue size, proportions and cell size in adult wings

The size of adult wings and their anterior and posterior compartments (in $\mu m^2$), wing proportions (width and length, in $\mu m$) and cell size (in $\mu m^2$) were all measured using Fiji Software (NIH, USA). Wing proportions (width and length) were measured along the red lines depicted in *Figure 2B*. The length corresponded to the distance between the tip of the anterior-posterior compartment boundary and the confluence of the anterior and posterior vein trunks in the hinge. The width corresponded to the distance between the anterior and posterior wing margins along a line orthogonal to the anterior posterior compartment boundary and depicted in the middle of the length's line. Cell size was measured as follows. Two conserved regions of a defined size between veins L4 and L5 (P compartment) and veins L2 and L3 (A compartment) were used to quantify the number of hairs (each wing cell differentiates a hair). Cell size was measured as the ratio between the size of the region and the number of hairs. The final values of *dpp-RNAi*-expressing wings were normalized as a percent of the control *gal4-driver; UAS-GFP-RNAi* values. At least 15 adult wings coming from different individuals were scored per genotype. Only adult males were scored. The average values and the corresponding standard deviations were calculated, and a Student t-test was carried out. Experimental conditions and control individuals driving the expression of control *UAS-GFP-RNAi* transgenes were grown in parallel. The AP boundary of adult *dpp-RNAi*-expressing wings in *Figure 2* was identified by the distinct pattern of bristles at the wing margin.

## Quantification of tissue growth in wing discs

The sizes of the wing disc and of the wing pouch were measured using Fiji Software (NIH, USA). Specified cell populations (notum, hinge and wing blade) are separated by epithelial folds, which are initiated by the apical shortening of cells at the early to mid-L3 stage. In order to delimit the wing pouch region, we always imaged apical sections of the wing disc and used the dorsal and ventral blade/hinge fold (according to DAPI staining) to delimit our wing pouch area. Only in *Figure 5*, we strictly followed the inner ring of Wg to delimit the wing pouch area. 25–75 wing discs were scored per genotype and experiment. The corresponding standard deviation was calculated, and a Student's t-test was carried out. Experimental conditions and individuals driving the expression of control UAS transgenes were grown in parallel. In *Figure 6*, after limiting the wing blade region of interest (represented by a dot line), quantification of the A, P, RFP-labelled, EGFP-labelled and non-expressing areas during the G-TRACE experiment were measured using Fiji Software (NIH, USA) manually (for the A and P) or by adjusting the signal threshold (RFP, EGFP). At least 20 wing discs per genotype and experiment were scored. The corresponding standard deviation was calculated, and the Student's t-test was carried out.

## Proliferation and growth rate measurements by clonal analysis in wing discs

Larvae from the following genotypes: (1) *hs-FLP, ubi-nls-RFP, FRT19A/FRT19A; tub-gal80ts/+; dpp-gal4/UAS-EGFP-RNAi* and (2) *hs-FLP, ubi-nls-RFP, FRT19A/FRT19A; tub-gal80ts/+; dpp-gal4/UAS-dpp-RNAi* were grown at 18°C, heat-shocked at 38°C for 45 min in early-second instar (96–108 hr AEL), returned to 18°C and transferred to 29°C in early third instar (156–168 hr AEL). Wing discs were dissected 42 hr thereafter, and the size of clones was quantified from confocal images with Fiji Software (NIH, USA). At least 80 clones in the wing blade and 35 clones in the hinge and body wall regions were quantified, and only clones/twin spots that did not fuse with neighboring clones/twin spots were used for the statistical analysis. Average values and the corresponding standard deviations were calculated, and a Student's t-test was carried out. Clones in the two different genotypes were induced and dissected always in parallel. Medial and lateral regions of control and *dpp*-depleted wing pouches were marked as depicted in *Figure 7—figure supplement 1*.

## Proliferation rate measurements by clonal analysis in pupal wings

Larvae from the following genotypes: (1) *hs-FLP; ubi-FRT-stop-FRT-GFP,tub-gal80ts; nub-gal4/UAS-dpp-RNAi* and *hs-FLP; ubi-FRT-stop-FRT-GFP/nub-gal4; tub-gal80ts/UAS-EGFP-RNAi* were grown at 18°C, heat-shocked at 38°C for 9 min in mid-third instar (8 days AEL) and transferred to 29°C. Those animals that entered puparium formation 24–30 hr thereafter were selected and pupal wings were dissected 24–30 hr after puparium formation (APF). At least 300 clones in the lateral and 400 clones in the medial regions from a minimum of 12 different pupal wings were analysed. The number of cells per clone was quantified from confocal images. Clones in the two different genotypes were induced and dissected in parallel. Medial and lateral regions of control and *dpp*-depleted pupal wings were marked as depicted in *Figure 7—figure supplement 1*.

## Statistical analysis

Standard deviation were calculated, and a Student's t-test was carried out in all cases with the help of Excel. $*p<0.05$; $**p<0.01$; $***p<0.001$. Graphical representations of data were done using Graph-Pad Prism version 6.07 or Microsoft Excel.

## Acknowledgements

We thank JP Vincent and K Basler for sharing unpublished results, A Ferreira for providing fly samples, R Barrio, S Cohen, J Colombani, C Estella, P Leopold, G Morata, G Pflugfelder, X Yang, the Bloomington *Drosophila* Stock Center (USA) and the Developmental Studies Hybridoma Bank (USA) for flies and antibodies, T Yates for text editing, and A Ferreira, L Murcia, C Recasens, C Santos and F Serras for comments on the manuscript and fruitful discussions. MM is an ICREA Research Professor. This work was funded by the *SIGNAGROWTH-BFU2013-44485* and *INTERGROWTH-BFU2016-77587-P* grants from MINECO (Government of Spain), and FEDER 'Una manera de hacer Europa'. We gratefully acknowledge institutional funding from the Spanish Ministry of Economy, Industry and Competitiveness (MINECO) through the Centres of Excellence Severo Ochoa award, and from the CERCA Programme of the Catalan Government.

## Additional information

### Funding

| Funder | Grant reference number | Author |
| --- | --- | --- |
| Ministerio de Economía y Competitividad | BFU2013-44485 | Marco Milan |
| Ministerio de Economía y Competitividad | BFU2016-77587-P | Marco Milan |

The funders had no role in study design, data collection and interpretation, or the decision to submit the work for publication.

### Author contributions

LB, Conceptualization, Data curation, Formal analysis, Validation, Investigation, Visualization, Methodology, Project administration, Writing—review and editing; MM, Conceptualization, Data curation, Formal analysis, Supervision, Funding acquisition, Validation, Visualization, Methodology, Writing—original draft, Project administration, Writing—review and editing

### Author ORCIDs

Marco Milán, http://orcid.org/0000-0002-7111-6444

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
