## [Decision Letter]

Thank you for submitting your article "Boundary Dpp promotes growth of medial and lateral regions of the *Drosophila* wing" for consideration by *eLife*. Your article has been reviewed by three peer reviewers, and the evaluation has been overseen by a Reviewing Editor and K VijayRaghavan as the Senior Editor. The reviewers have opted to remain anonymous.

We do sincerely apologise for the delay in sending you this decision. The reviewers have discussed the reviews with one another and the Reviewing Editor has drafted this decision to help you prepare a revised submission.

Summary:

A contentious item that continues to raise interest concerns the relationship between the gradient of the BMP4-like signaling protein Dpp produced in *Drosophila* wing discs and cell proliferation in the disc. Dpp is both necessary and sufficient for disc growth, and the problem basically boils down to why regions with different levels of Dpp and BMP signaling do not cause different amounts of growth. Evidence and arguments on this point remain of high current interest.

The present manuscript is a partial rebuttal to the 2015 Nature paper from Akiyama and Gibson, which used various types of dpp loss-of-function clones to argue that the BMP Dpp produced by the stripe of cells anterior to the A/P boundary in wing discs was not necessary for the growth of the disc. This argued that models based on reading a gradient of Dpp were likely wrong, and that levels could be greatly reduced without greatly affecting growth, and thus models based on a temporal gradient of increasing BMP signaling were probably wrong as well. Nonetheless, that study showed that the Dpp produced by the entire anterior compartment was necessary for growth, at least up until 36 hours before wandering third instar, presumably this was supplied by low-level Dpp produced outside the normal stripe of high level Dpp expression.

The present manuscript argues that stripe Dpp is necessary for growth both at early and later time points. However, the authors propose that Dpp levels are more permissive than instructive for growth, which is at least partly in line with the Akiyama study.

Essential revisions:

Reviewer #1:

1) The authors need to do a better job explaining the technical differences that they think account for their different results. As is, there are two rather confusing sentences at the end of the Discussion, which no one will understand who has not read Akiyama. The two studies use the same dpp-gal4 line, as far as I can tell, to limit Dpp-loss to the stripe. Rather, the differences may lie in the regions being measured, and in the speed with which Dpp is eliminated by the two techniques. Akiyama measured total disc size, where the present study sees stronger effects by additionally measuring just the wing pouch region. The present manuscript uses UAS-RNAi expression, while Akiyama used GAL4-driven mutation of a conditional dpp allele. Arguably, the RNAi will more quickly remove RNA, while loss of the dpp gene will leave in place RNA that perdures.

It would also be helpful if the authors looked at when Dpp was lost and pMad disrupted in early stage discs, as Akiyama did. It should be noted that Akiyama can see loss of Dpp and pMad over especially the dorsal part of the pouch as early as 72 hours AEL, without that obviously altering subsequent growth of the dorsal pouch (although specific measurements were lacking). So I suspect that the different result is due to how much residual signaling is left over, and perhaps how and when that affects brinker. Confirming that would require, however, a careful comparison of signaling levels using the two techniques, or at least of brinker expression at an early stage, something not done here.

2) While the authors use multiple RNAi lines for their initial studies on general removal of Dpp, they only show data with one RNAi line for the critical experiment removing stripe Dpp with dpp-gal4. They need to show similar results with a second independent line, as size reductions can result from off-target effects of RNAI. This is especially important because of the surprising result that size reduction is stronger in the anterior compartment, where the RNAi is being expressed.

3) Both Akiyama and the present study do an analysis of the effects of timed, general Dpp removal of Dpp. The authors should do a better job explaining what is different with their results. Both agree that early general loss causes growth defects, something that the authors do not, I think, mention. The main difference seems to be the late effects shown in the new manuscript.

4) Akiyama claims there is a critical difference between removing Dpp from the stripe and removing Dpp from the entire anterior compartment. Thus, it is important to know where Dpp is being removed using dpp-gal4 and dpp-RNAi; the anterior boundary of dpp-gal4 expression is not strict, and as the authors note also shifts during development; the dpp-gal4 may as well. The authors try to answer this using G-TRACE to label cells whose ancestors expressed dpp-gal4. They state: "However, this domain never covered the entire A compartment, 193 especially in those wing discs carrying the dpp-RNAi transgene". First, that is not quite true in the dorsal part of the pouch, where coverage is pretty complete with the 48-hour shift (and similar to the figure in Evans et al.). Second, it is not clear from the figure that this really is different with the dpp-RNAi, nor is it clear why it should be different. And finally, although the G-TRACE domain is much narrower when FLPase expression is limited to later stages, I think some caution is still in order. In my experience the extent of FLPout labeling depends a lot on the FLPase-FLPout combination being used, suggesting that there is often residual GAL4 expressed that does not cause FLPout of all reporters. Thus, it remains a formal possibility that the dpp-GAL4 driven RNAi is knocking down some low level of Dpp made outside the stripe, and I think this should be acknowledged.

5) Since the authors believe that stripe Dpp is important for growth, it is not clear how this shows that Dpp signaling is permissive. While it is (still) true that growth in not higher where Dpp is higher, the finding that the stripe is necessary brings back into play models using gradient steepness or temporal changes in signaling levels. The authors talk about spread of Dpp, but not clearly enough to explain how this works. Is there a region of the pouch that they think is not seeing sufficient Dpp for growth? Then why is it still growing? This needs to be clarified.

If the added data with brinker knockdown pertains, that also needs to be incorporated into the discussion. The authors might also want to briefly discuss the new paper from the Irvine lab, which suggests that growth can be a little higher in the dpp stripe if mechanical feedback, presumably different at the edges of the disc, is altered.

Reviewer #2:

In this article, the authors have tried to address the temporal requirement of Dpp for growth and patterning of the *Drosophila* wing. RNAi mediated depletion of Dpp using three GAL4 drivers (nubbin GAL4, dpp GAL4 and ptc GAL4) recombined with tub-GAL80ts is the methodology utilized to temporally deplete Dpp. From these experiments, the authors conclude that Dpp is continuously required throughout the third instar larval stage to initiate cell proliferation and to control the final tissue size. The quantification of sizes and sampling of large data sets (as seen in the scatter plot in Figure 4) is commendable as it very well exemplifies the intrinsic differences in wing disc size (with n ~ 40-70 discs) and also sums up the temporal effect of "dpp RNAi" on wing disc size.

This conclusion is in contrast to the conclusion from Akiyama and Gibson, 2015 wherein conditional removal of genomic Dpp was achieved by using CRISPR to insert FRT sites across the first coding exon of Dpp and FLP was expressed using a number of GAL4 drivers (dpp GAL4, ci GAL4, en GAL4, ap GAL4, nub GAL4). They observed that expression of FLP (thus removal of a part of genomic Dpp) using dpp GAL4 or en GAL4 didn't alter the final size of wing disc while expressing FLP using ci GAL4, nub GAL4 or ap GAL4 affected the final size of the wing disc consistent with their interpretation that Dpp produced by anterior cells but not the compartmental stripe itself is essential for growth. They have also addressed the temporal requirement of Dpp by recombining tub-GAL80ts with ci GAL4 and conclude that Dpp is indeed required throughout the third instar larval stage to promote growth.

These two results essentially emphasize the central dogma of life: DNA is transcribed to mRNA which is then translated into a protein. There are a number of layers of regulation in each of these cellular processes (promoters and enhancers controlling the transcription rate, mRNA production and degradation rates, protein production and degradation rates, formation of functional protein from precursor, protein localization to correct compartments, etc.), which can affect the activity of the protein. The claim in the Discussion of this article reads as below: "On the basis of our results, we would like to propose that the stability of the gene product and percentage of cells in which the gene product has been efficiently removed in the two chromosomes are parameters that should be taken into consideration to interpreting the phenotypic consequences of site-specific recombination experiments and when generating conditional null alleles". The major caveat to warrant such a proposal lies in the fact that throughout the article depletion of Dpp is indirectly read via immunostaining of downstream targets of Dpp (Spalt/Omb/Brinker) or via interpreting the final wing morphology. There is no direct quantification of reduction of levels of Dpp (mRNA or Protein) and there is no rescue experiment to suggest that the differences observed is indeed due to depletion of Dpp. Although 5 different RNAi were used (Figure 1 to D), all 5 are against the (non-coding) Exon 1 of dpp mRNA. It will be imporrtant to show the levels of depletion of Dpp in all the temporal experiments.

*Reviewer #3:*

The mechanisms controlling tissue growth during animal development are not well understood. Classical wing disc-specific loss-of-function alleles of Dpp (dpp-disk alleles) identified a role of Dpp in controlling tissue growth in the *Drosophila* wing. Those alleles remove Dpp activity constitutively throughout development of the wing and therefore do not provide information on temporal requirements of Dpp signaling for growth. Recent work using inducible systems suggested that Dpp signaling is only required early in wing development to sustain tissue growth, but is dispensable later on. This manuscript by Milan and colleague carefully addresses this question. The high quality data they provide convincingly show that Dpp signaling is required throughout wing disc development to permit the growth and proliferation of wing disc cells throughout the entire wing pouch primordium.

---

## [Author Response]

*Essential revisions:*

*Reviewer #1:*

*1) The authors need to do a better job explaining the technical differences that they think account for their different results. As is, there are two rather confusing sentences at the end of the Discussion, which no one will understand who has not read Akiyama.*

We appreciate this particular comment.

We have reformatted the whole paragraph at the end of the Discussion, as shown below, to make it understandable to those that have not read Akiyama’s paper.

“The Cre-Lox and FLP-FRT recombination systems are widely used in developmental biology to induce conditional null alleles and address gene function. […] Alternatively, the FLP-FRT recombination system used in (Akiyama and Gibson, 2015) might not have been able to efficiently remove gene activity in the two chromosomes in all boundary cells.”

The two studies use the same dpp-gal4 line, as far as I can tell, to limit Dpp-loss to the stripe.

In our work, we have used the *dpp(disk)-gal4* driver described in (Staehling-Hampton et al., 1994) that contains the 4-kb DNA fragment within the 3’disk region called the blk enhancer fragment. We have chosen this gal4 driver as it is expressed in all *dpp*-producing cells in imaginal tissues (Masucci et al., 1990) and is known to rescue *dpp-disk* alleles when driving Dpp expression (Staehling-Hampton et al., 1994). According to the Methods’ section in Akiyama and Gibson, 2015, authors also used the *dpp(disk)-gal4* driver described in (Staehling-Hampton et al., 1994).

*Rather, the differences may lie in the regions being measured, and in the speed with which Dpp is eliminated by the two techniques.*

As shown in Figure 3 of our paper, a 24-hours induction of dpp-RNAi expression is required to induce robust loss of pMAD and gain of Brinker expression in all wing pouch cells. We have now monitored dpp mRNA levels in our temperature shift experiments, and show that mRNA levels are already lower after a period of 12-24 h of induction but longer exposures are being required to induce robust (though not complete) reduction of dpp mRNA levels. Data are now included in Figure 3—figure supplement 1 and discussed in subsection “Dpp emanating from the AP boundary is continuously required for growth of the wing pouch” as follows:

“We first characterised the kinetics of Dpp depletion by shifting the larvae from 18ºC to 29ºC at different time points of the third instar stage […] These results indicate that the wing pouch levels of Brk and pMAD proteins are highly sensitive to reductions in Dpp, and that Spalt and Omb either require stronger or longer reduction in Dpp activity levels or they are highly stable proteins, which might contribute to the robustness of Dpp-mediated patterning of the wing.”

Unfortunately, Akiyama and Gibson did not carry out a careful analysis of when Dpp was removed and when Dpp activity levels started to be affected. They just analyzed Dpp protein expression and activity (pMad) at different time points of development, but without controlling FLP expression. In their case, stability of the dpp mRNA might increase even further the perdurance of gene function thus contributing to the lack of growth phenotype.

*Akiyama measured total disc size, where the present study sees stronger effects by additionally measuring just the wing pouch region.*

We have measured the impact on the size of the wing pouch (presumptive wing blade) and on the whole wing disc upon RNAi-mediated depletion of *dpp* expression at the AP boundary. As shown in Figure 4 and Figure 5, the impact on the size of the wing pouch is much stronger than on the whole wing disc, reflecting the wing pouch-specific growth promoting activity of Dpp.

Unfortunately, Akiyama and Gibson, 2015 provides only wing disc measurements, and subtle reductions in wing pouch size might have not been detected.

*The present manuscript uses UAS-RNAi expression, while Akiyama used GAL4-driven mutation of a conditional dpp allele. Arguably, the RNAi will more quickly remove RNA, while loss of the dpp gene will leave in place RNA that perdures.*

As explained above, we have monitored the kinetics of *dpp* mRNA depletion upon temporally controlled expression of *dpp-RNAi* in the *dpp-gal4* domain (see Figure 3—figure supplement 1). While 12 to 24 hours of induction are sufficient to induce visible depletion of *dpp* mRNA, longer induction protocols are required to get a robust reduction in *dpp* mRNA levels. As stated by the reviewer, stability of the *dpp-RNA* might increase even further the delay in achieving sufficient depletion of dpp mRNA levels in the FRT-experiments carried out in Akiyama and Gibson, 2015.

*It would also be helpful if the authors looked at when Dpp was lost and pMad disrupted in early stage discs, as Akiyama did. It should be noted that Akiyama can see loss of Dpp and pMad over especially the dorsal part of the pouch as early as 72 hours AEL, without that obviously altering subsequent growth of the dorsal pouch (although specific measurements were lacking). So I suspect that the different result is due to how much residual signaling is left over, and perhaps how and when that affects brinker. Confirming that would require, however, a careful comparison of signaling levels using the two techniques, or at least of brinker expression at an early stage, something not done here.*

We agree with reviewer’s observation that specific measurements were lacking in Akiyama and Gibson, 2015. Indeed, and as indicated above, Akiyama’s paper only provides wing disc measurements, and subtle reductions in wing pouch size might have not been detected.

As suggested by the reviewer, we have looked at when pMAD is disrupted and Brinker is gained in the wing pouch and wing disc of early stage larvae expressing *dpp-RNAi* under the control of the *dpp-gal4* driver. These new data are included as Figure 5—figure supplement 1 and show (1) that Dpp activity levels are strongly reduced in early third instar wing discs (72 h AEL) expressing dpp-RNAi in a chronic manner and (2) that Dpp activity levels are already reduced in early third instar wing discs subjected to depletion of *dpp* for 12 h of induction and continue to be reduced upon 24 and 36 hours of induction. These new data nicely complement the impact of *dpp* depletion on the growth rates of early wing discs shown in Figure 5 and present evidence that the RNAi system is more efficient in depleting Dpp activity than the FRT-mediated system used in Akiyama’s paper. As noted by the Reviewer, pMAD was not completely removed from the wing pouch of early wing discs subjected to FRT-mediated depletion of Dpp in Akiyama’s paper (Figure 3). We appreciate this observation as this might explain their lack of growth phenotype.

*2) While the authors use multiple RNAi lines for their initial studies on general removal of Dpp, they only show data with one RNAi line for the critical experiment removing stripe Dpp with dpp-gal4. They need to show similar results with a second independent line, as size reductions can result from off-target effects of RNAI. This is especially important because of the surprising result that size reduction is stronger in the anterior compartment, where the RNAi is being expressed.*

As suggested by the reviewer, we have now used three independent dpp-RNAi lines to deplete stripe Dpp with the dpp-Gal4 driver. Data are now included as Figure 4—figure supplement 3 and indicate (1) that the effect on the size of the A compartment is always stronger than on the size of the P compartment, and (2) that the effect on the size of the wing pouch is always stronger than the effect on the size of the wing disc. These new data are now discussed in the paper (subsection “Dpp emanating from the AP boundary is continuously required for growth of the wing pouch”) as follows:

“The size of the A compartment was also reduced to a larger extent than the size of the P compartment (Figure 4—figure supplement 3). Similar results were obtained with other dpp-RNAi lines (Figure 4—figure supplement 3).”

*3) Both Akiyama and the present study do an analysis of the effects of timed, general Dpp removal of Dpp. The authors should do a better job explaining what is different with their results. Both agree that early general loss causes growth defects, something that the authors do not, I think, mention. The main difference seems to be the late effects shown in the new manuscript.*

We have summarized, in the Introduction, the major conclusions of Akiyama and Gibson, 2005’s paper, especially those concerning the temporal growth-promoting requirement of Dpp:

Introduction: “CRISPR-Cas9-mediated genome editing of the dpp locus combined with the FLP-FRT recombination system to temporally control the removal of Dpp from its endogenous stripe domain has come to the conclusion that Dpp emanating from the AP boundary is not required for wing growth and that the growth-promoting role of Dpp is restricted to the early stages of wing development (Akiyama and Gibson, 2015).”

We now include in the last paragraph of our paper the following sentences:

“The recent use of an FRT-dependent conditional null allele of dpp came to the unexpected conclusion that Dpp emanating from the AP boundary is not required for wing growth and that the growth-promoting role of Dpp is restricted to the early stages of wing development (Akiyama and Gibson, 2015).”

Concerning the spatial requirement of Dpp in promoting growth, Akiyama and Gibson conclude, as stated in the last paragraph of their paper, that “[…] abrogation of the compartment-boundary-centred Dpp signaling gradient did not disrupt active cell proliferation and elicited only mild growth defects during the third larval instar”. This is in conflict with all our data, as stated now in the last paragraph of our paper, “our results, based on the use of RNAi hairpins targeting dpp, do not validate their conclusions on the […]spatial requirement of Dpp expression and their impact on growth.”

Concerning the early requirement of Dpp in promoting growth, Akiyama and Gibson conclude, as stated in the last paragraph of their paper, that “[…] eliminating Dpp throughout the disc at early larval stages caused severe growth defects”. Concerning the late requirement of Dpp in promoting growth, Akiyama and Gibson conclude, as stated in the last paragraph of their paper, that “Dpp […]is not continuously required to drive proliferative growth in the latter half of larval development”. This is in conflict with all our data, as stated now in the last paragraph of our paper, “our results, based on the use of RNAi hairpins targeting dpp, do not validate their conclusions on the temporal […] requirement of Dpp expression and their impact on growth.”

*4) Akiyama claims there is a critical difference between removing Dpp from the stripe and removing Dpp from the entire anterior compartment. Thus, it is important to know where Dpp is being removed using dpp-gal4 and dpp-RNAi; the anterior boundary of dpp-gal4 expression is not strict, and as the authors note also shifts during development; the dpp-gal4 may as well. The authors try to answer this using G-TRACE to label cells whose ancestors expressed dpp-gal4. They state: "However, this domain never covered the entire A compartment, 193 especially in those wing discs carrying the dpp-RNAi transgene". First, that is not quite true in the dorsal part of the pouch, where coverage is pretty complete with the 48-hour shift (and similar to the figure in Evans et al.). Second, it is not clear from the figure that this really is different with the dpp-RNAi, nor is it clear why it should be different.*

We agree with reviewer’s comments and have deleted the previous conclusion. The text remains as follows (subsection “Dpp emanating from the AP boundary is continuously required for growth of the wing pouch”):

“In the case of induction periods of 24 h or 36 h, this domain occupied a small fraction of the central region of the wing primordium, and the non-autonomous effects on tissue size were significant, visualised by the impact on the size of the P compartment and on the size of the A compartment not labelled by EGFP (Figure 6).”

*And finally, although the G-TRACE domain is much narrower when FLPase expression is limited to later stages, I think some caution is still in order. In my experience the extent of FLPout labeling depends a lot on the FLPase-FLPout combination being used, suggesting that there is often residual GAL4 expressed that does not cause FLPout of all reporters. Thus, it remains a formal possibility that the dpp-GAL4 driven RNAi is knocking down some low level of Dpp made outside the stripe, and I think this should be acknowledged.*

We agree with the reviewer about re-phrasing the conclusions from the G-trace experiments. Gibson’s paper proposes that low levels outside the AP boundary can promote growth in the absence of boundary Dpp. In order to address this particular point, we have included the new data on dpp mRNA expression upon temporally controlled boundary expression of dpp-RNAi in the context of the G-trace experiments. The whole paragraph reads as follows (subsection “Dpp emanating from the AP boundary is continuously required for growth of the wing pouch”):

“Although it is still possible that some low level of Dpp made outside the stripe might be knocked down in our experiments, […] conflict with the previously proposed role of low levels of non-boundary Dpp in promoting wing growth (Akiyama and Gibson, 2015), and it strongly suggests that wing growth requires high levels of boundary Dpp.”

*5) Since the authors believe that stripe Dpp is important for growth, it is not clear how this shows that Dpp signaling is permissive. While it is (still) true that growth in not higher where Dpp is higher, the finding that the stripe is necessary brings back into play models using gradient steepness or temporal changes in signaling levels. The authors talk about spread of Dpp, but not clearly enough to explain how this works. Is there a region of the pouch that they think is not seeing sufficient Dpp for growth? Then why is it still growing? This needs to be clarified.*

*If the added data with brinker knockdown pertains, that also needs to be incorporated into the discussion.*

We have now included the new data on brinker knockdown to reinforce the proposal that Dpp plays a permissive role in regulating growth simply by maintaining the levels of Brinker below a growth-repressing threshold. Data are now included as Figure 8 and Figure 8—figure supplement 1, and discussed as a new section at the end of the Results. We have used two different Gal4 drivers (nub-Gal4 and rotund-Gal4) expressed in the wing pouch region to carry out the double knockdown. In both cases, the tissue size rescue (but not the patterning) is evident. We noticed that some of the genotypes in our first version of Figure 8 were mislabeled but these have been now corrected.

We have also reformatted the first half of the second paragraph of the Discussion section to incorporate the Brinker data and to discuss the implications on the “steepness” or “temporal-rule” models:

“Our observations that co-depletion of Dpp and Brinker gives rise to nearly wild type-sized adult wings and that Dpp depletion has an equal impact on the growth rates of medial and lateral regions of the developing wing indicate that graded activity of Dpp is not an absolute requirement for growth and that Dpp plays a permissive role in regulating proliferative growth by maintaining Brk expression levels below a growth-repressive threshold. […] and the more recent “temporal rule” model that postulates that cells divide when Dpp signaling levels have increased by 50% (Wartlick et al., 2011).”

We have previously shown in Ferreira and Milán, 2015 that changes in the levels of Dally can have an impact on the growth and proliferation rates of the wing. Thus, overexpression of Dally increases Dpp spreading (without changing the absolute amount of Dpp signaling in the tissue) and gives rise to bigger wings. On the contrary, downregulation of Dally reduces Dpp spreading and causes tissue undergrowth. These data indicate that it is not the amount of Dpp but how far it can travel what regulates the final size of the wing. We have included these data as Figure 8—figure supplement 2 and reformatted the second half of the second paragraph of the Discussion section to incorporate a clearer explanation of the role of Dpp spreading in regulating wing size:

“How is then the size of the wing controlled by Dpp? Of remarkable interest is the capacity of overexpression of Dally, a proteoglycan that contributes to Dpp stability (Akiyama et al., 2008), […] These results support the proposal that the range of Dpp spreading emanating from the AP boundary can regulate, in an instructive manner, the final size of the developing wing appendage.”

*The authors might also want to briefly discuss the new paper from the Irvine lab, which suggests that growth can be a little higher in the dpp stripe if mechanical feedback, presumably different at the edges of the disc, is altered.*

This is a good point. However, we rather prefer not to discuss this point as the Dpp field is already quite complex (different models, conflicting interpretations, etc), and we prefer to maintain the message simple and to the point. I hope Reviewer will understand.

*Reviewer #2:*

*In this article, the authors have tried to address the temporal requirement of Dpp for growth and patterning of the Drosophila wing. RNAi mediated depletion of Dpp using three GAL4 drivers (nubbin GAL4, dpp GAL4 and ptc GAL4) recombined with tub-GAL80ts is the methodology utilized to temporally deplete Dpp. From these experiments, the authors conclude that Dpp is continuously required throughout the third instar larval stage to initiate cell proliferation and to control the final tissue size. The quantification of sizes and sampling of large data sets (as seen in the scatter plot in Figure 4) is commendable as it very well exemplifies the intrinsic differences in wing disc size (with n ~ 40-70 discs) and also sums up the temporal effect of "dpp RNAi" on wing disc size.*

*This conclusion is in contrast to the conclusion from Akiyama and Gibson, 2015 wherein conditional removal of genomic Dpp was achieved by using CRISPR to insert FRT sites across the first coding exon of Dpp and FLP was expressed using a number of GAL4 drivers (dpp GAL4, ci GAL4, en GAL4, ap GAL4, nub GAL4). They observed that expression of FLP (thus removal of a part of genomic Dpp) using dpp GAL4 or en GAL4 didn't alter the final size of wing disc while expressing FLP using ci GAL4, nub GAL4 or ap GAL4 affected the final size of the wing disc consistent with their interpretation that Dpp produced by anterior cells but not the compartmental stripe itself is essential for growth. They have also addressed the temporal requirement of Dpp by recombining tub-GAL80ts with ci GAL4 and conclude that Dpp is indeed required throughout the third instar larval stage to promote growth.*

*These two results essentially emphasize the central dogma of life: DNA is transcribed to mRNA which is then translated into a protein. There are a number of layers of regulation in each of these cellular processes (promoters and enhancers controlling the transcription rate, mRNA production and degradation rates, protein production and degradation rates, formation of functional protein from precursor, protein localization to correct compartments, etc.), which can affect the activity of the protein. The claim in the Discussion of this article reads as below: "On the basis of our results, we would like to propose that the stability of the gene product and percentage of cells in which the gene product has been efficiently removed in the two chromosomes are parameters that should be taken into consideration to interpreting the phenotypic consequences of site-specific recombination experiments and when generating conditional null alleles". The major caveat to warrant such a proposal lies in the fact that throughout the article depletion of Dpp is indirectly read via immunostaining of downstream targets of Dpp (Spalt/Omb/Brinker) or via interpreting the final wing morphology. There is no direct quantification of reduction of levels of Dpp (mRNA or Protein) and there is no rescue experiment to suggest that the differences observed is indeed due to depletion of Dpp. Although 5 different RNAi were used (Figure 1 to D), all 5 are against the (non-coding) Exon 1 of dpp mRNA.*

As stated in the first sentence of the Results section, our “five RNAi hairpins, one long and four short target various regions of the first dpp coding exon (Perkins et al., 2015)”.

Since all five RNAi lines were only used in Figure 1 with the nub-gal4 driver, we have now used three independent dpp-RNAi lines to deplete Dpp also with the dpp-Gal4 driver. Data are now included as Figure 4—figure supplement 3 and we observed the same effects on tissue size.

*It will be imporrtant to show the levels of depletion of Dpp in all the temporal experiments.*

We have monitored the expression of dpp mRNA by in situ hybridization in wing discs subject to boundary expression of dpp-RNAi for 12, 24, 36 and 48 hours. We have used as controls wing discs of the same genotype but raised at 18ºC. All samples were processed in parallel. Data are now included in Figure 3—figure supplement 1 and discussed in subsection “Dpp emanating from the AP boundary is continuously required for growth of the wing pouch”:

“We first characterised the kinetics of Dpp depletion by shifting the larvae from 18ºC to 29ºC at different time points of the third instar stage (Figure 3) and analysing dpp expression and Dpp activity levels in late third instar wing discs. […] These results indicate that the wing pouch levels of Brk and pMAD proteins are highly sensitive to reductions in Dpp, and that Spalt and Omb either require stronger or longer reduction in Dpp activity levels or they are highly stable proteins, which might contribute to the robustness of Dpp-mediated patterning of the wing “

We have also monitored the expression of dpp mRNA by in situ hybridization in wing discs subjected to chronic expression of dpp-RNAi in the wing pouch region (with the nub-gal4 driver). We have used as controls wing discs expressing GFP. Both genotypes were processed in parallel. Data are now included in Figure 1 and discussed in subsection “Independent RNAi hairpins to deplete Dpp in the developing wing” and legend to Figure 1, as follows:

“We next characterised the effects of dpp depletion on the size of the developing wing appendage and monitored Dpp expression and activity levels. […]”

“As expected, dpp mRNA levels […] decreased in the developing wing appendage (Figure 1).”

Legend to Figure 1: “[…] dpp mRNA levels are reduced in the wing pouch (wp) when compared to the hinge region (black arrows).”